# Learning in Context, Guided by Choice: A Reward-Free Paradigm for Reinforcement Learning with Transformers

## Abstract

In-context reinforcement learning (ICRL) leverages the in-context learning capabilities of transformer models (TMs) to efficiently generalize to unseen sequential decision-making tasks without parameter updates. However, existing ICRL methods require explicit reward signals during pretraining, limiting their applicability in real-world scenarios where rewards are ambiguous, difficult to specify, or expensive to collect. To overcome this limitation, we propose a new learning paradigm, *In-Context Preference-based Reinforcement Learning* (ICPRL), where both the pretraining of TMs and their deployment to new tasks rely solely on preference data, thereby eliminating the need for reward supervision. We study two variants that differ in the granularity of feedback: *Immediate Preference-based RL* (I-PRL) with per-step preferences, and *Trajectory Preference-based RL* (T-PRL) with trajectory-level comparisons. We first show that supervised pretraining, a proven strategy in ICRL, remains effective in ICPRL for training TMs to predict optimal actions using preference-based context datasets. To improve data efficiency, we further propose alternative frameworks for I-PRL and T-PRL that directly optimize TM policies from preference data without relying on optimal action labels or reward signals. Empirical evaluations on dueling bandits, navigation, and continuous control tasks demonstrate that ICPRL enables strong generalization to unseen RL tasks, achieving performance on par with ICRL methods trained with full reward supervision.

## 1 Introduction

Reinforcement learning (RL) has achieved impressive successes across a wide range of domains, including robotics (Kober et al., 2013), recommendation systems (Afsar et al., 2022), and more recently, post-training of large language models (Ziegler et al., 2019). Despite these advances, RL remains data-intensive: both online and offline RL methods typically require a large number of environment interactions to achieve satisfactory performance. To address this issue, *in-context reinforcement learning* (ICRL) has recently gained attention (Laskin et al., 2022). Building on the in-context learning capabilities of transformer models (TMs), ICRL methods pretrain TMs on a diverse set of RL tasks and deploy them directly to *new* tasks at test time. These pretrained TMs act as meta-policies, rapidly adapting to unseen tasks using only tens of trajectories and without any parameter updates. See Figure 1 for visuals. Notably, ICRL achieves strong generalization and high performance with significantly fewer environment interactions than conventional RL learners (Lee et al., 2024; Dong et al., 2024).

**Challenges.** Existing ICRL methods, however, rely on access to *explicit reward signals* during pretraining. This limits their applicability in many real-world scenarios where rewards are ambiguous, difficult to specify, or expensive to collect. While reward design is already a major challenge in standard RL (Christiano et al., 2017; Ibarz et al., 2018), ICRL exacerbates the problem, as reward signals must be meaningful and consistent *across* tasks to enable generalization.

**Contributions.** To overcome this limitation, we propose a new learning paradigm: *In-Context Preference-based Reinforcement Learning* (ICPRL), where both pretraining and deployment rely solely on preference feedback, eliminating reward supervision. Our main contributions are:

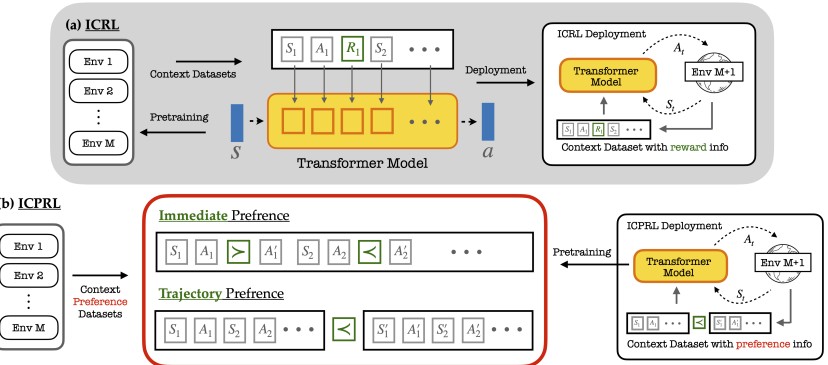

Figure 1: **(a)** ICRL methods adapt to *new* RL tasks using in-context learning, but both their pretraining and deployment require access to reward signals, which can be costly or impractical to obtain in many settings. **(b)** This work proposes a novel ICRL paradigm that uses *only preference data* for both pretraining and deployment, eliminating the need for explicit reward supervision.

- We introduce two variants of ICPRL that differ in the *granularity* of preference feedback: *Immediate Preference-based RL* (I-PRL), which uses per-step preferences, and *Trajectory Preference-based RL* (T-PRL), which relies on comparisons between full trajectories. Figure 1 illustrates the proposed ICPRL paradigms.

- For both settings, we first propose a simple *supervised method* that trains a TM to predict optimal actions for query states, conditioned on context datasets constructed from preference data. We show that supervised pretraining, an established approach in ICRL, is also effective for ICPRL.

- Recognizing the cost of obtaining high-quality action labels for supervision, we further propose two frameworks, one for I-PRL and one for T-PRL, that more effectively leverage the preference data during pretraining, eliminating the need for *both* reward signals and optimal action labels.

**Empirical Results.** We evaluate our frameworks on a suite of challenging tasks, including dueling bandits (Yue et al., 2012), navigation (Laskin et al., 2022), and continuous control (Yu et al., 2020). Remarkably, despite being pretrained using only preference data, our transformer models exhibit strong generalization to unseen RL tasks, often even matching the performance of ICRL baselines pretrained with full reward supervision.

## 2  RELATED WORK

We discuss the most relevant related work, deferring the full literature review to Appendix B .

**Preference-based Reinforcement Learning (PbRL).** PbRL focuses on learning from preference signals, typically given as comparisons between actions or trajectories, instead of relying on scalar reward functions (Christiano et al., 2017; Wirth et al., 2017; Brown et al., 2019). A common paradigm in PbRL involves a two-stage model-based process: first, learning a reward function from preferences, and second, optimizing a policy using standard RL algorithms with the learned reward (Ibarz et al., 2018; Lee et al., 2021; Liu et al., 2023). The most salient distinction between the proposed ICPRL paradigm and prior PbRL approaches is that PbRL assumes the *same* environment for training and evaluation, whereas ICPRL operates in the in-context setting, where a model must generalize to *new* tasks at inference time without parameter updates.

**Transformer Models and In-Context Reinforcement Learning.** Transformer models have demonstrated superior performance in RL problem (Li et al., 2023; Yuan et al., 2023). Building on the in-context learning abilities of transformers, ICRL methods aim to learn a TM-based meta-policy to generalize to unseen tasks in context. ICRL methods differ in their requirements of context datasets. For example, *Algorithm Distillation* (AD) (Laskin et al., 2022) and its variations (Zisman et al., 2023; Tarasov et al., 2025) uses sequential modeling to emulate the learning process of RL algorithms, i.e., meta-learning (Vilalta & Drissi, 2002). Decision transformer (Chen et al., 2021)(DT)-based methods rely on return-to-go to guide the transformer models to generalize to new tasks (Grigsby et al., 2023;

Huang et al., 2024; Schmied et al., 2024). A recent framework *Decision Pretrained Transformer* (DPT) uses supervised-pretraining for in-context decision making. DPT trains transformers to predict the optimal action given a query state and a set of transitions. Despite its strong performance, DPT requires high-quality action labels for its pretraining. However, to our best knowledge, all existing ICRL methods assume explicit reward (goal) signal for pretraining. To this end, we make the first step towards reward/goal-free ICRL.

## 3 PRELIMINARY

**Markov Decision Process (MDPs).** We consider standard episodic RL problem, defined as MDPs. An MDP $\tau$ is a tuple $(\mathcal{S}, \mathcal{A}, P_\tau, R_\tau, \rho_\tau, H)$ where $\mathcal{S}$ and $\mathcal{A}$ are the sets of all possible states and actions, $P_\tau : \mathcal{S} \times \mathcal{A} \to \Delta(\mathcal{S})$ is the distribution of the next state, $R_\tau : \mathcal{S} \times \mathcal{A} \to \mathbb{R}$ is the reward function, $\rho_\tau \in \Delta(\mathcal{S})$ is the initial state distribution, and $H$ is the episode horizon. At the initial step $h = 1$, an initial state $s_1 \in \mathcal{S}$ is sampled according to $\rho_\tau$. At each time step $h$, the agent chooses action $a_h \in \mathcal{A}$ and receives reward $r_h = R_\tau(s_h, a_h)$. Then the next state $s_{h+1}$ is generated following $P_\tau(s_h, a_h)$. A policy $\pi : \mathcal{S} \to \Delta(\mathcal{A})$ maps the current state to an action distribution. Let $G_\tau(\pi) = \mathbb{E}[\sum_{h=1}^H r_h | \pi, \tau]$ denote the expected cumulative reward of $\pi$ for task $\tau$. The goal of an agent is to learn the optimal policy $\pi_\tau^\star$ that maximizes $G_\tau(\pi)$.

**ICRL.** ICRL methods first pretrain a transformer $T_\theta$ on diverse RL tasks $\{\tau_i\}_{i=1}^m$, where each $\tau_i \sim p_\tau$ is an MDP drawn independently from the task distribution $p_\tau$. The pretrained policy $T_\theta$ is then deployed on a new task $\tau^{\text{test}} \sim p_\tau$[1]. The **goal** is to pretrain $T_\theta$ so it generalizes *in context* to $\tau^{\text{test}}$ during deployment. To facilitate in context generalization, ICRL provides a **context dataset** $D^R$ with *reward information* about $\tau^{\text{test}}$, and the policy $T_\theta(a|s, D^R)$ outputs actions $a$ for given states $s$, conditioned on $D^R$. In other words, $T_\theta$ acts as a meta-policy, adapting to tasks conditioned on the context datasets collected from these tasks. Different ICRL methods assume different dataset formats. For example, DPT assumes $D^R = \{s_j, a_j, r_j, s_j'\}_{j \in [H]}$, a set of transitions collected by a behavior policy (Lee et al., 2024); policy distillation methods like AD assume $D^R = \{\xi_{i,j}\}_{j=1}^J$, a set of trajectories $\xi_{i,j}$ collected by increasingly improving policies (Laskin et al., 2022). Despite these format differences, existing ICRL methods typically assume that context datasets include rewards.

**Supervised Pretraining for ICRL.** One representative method for ICRL is DPT, which is the pioneering work that proposes to use supervised pretraining for ICRL. DPT assumes a pretraining dataset $\mathcal{D}_i = \{s_i^{\text{query}}, a_i^\star, D_i^R\}$ for each pretraining task $\tau_i$ where $D_i^R$ is the context dataset introduced above, $s_i^{\text{query}}$ is a query state to be used for supervised pretraining, and $a_i^\star \sim \pi_{\tau_i}^\star(s_i^{\text{query}})$ is an associated optimal action label for $s_i^{\text{query}}$ sampled from the optimal policy $\pi_{\tau_i}^\star$ for $\tau_i$ (Lee et al., 2024). DPT simply pretrains the TM policy $T_\theta$ to predict the optimal actions for query states given the context datasets with the following classification-like objective:

$$\max_\theta \frac{1}{m} \sum_{i=1}^m \log T_\theta(a_i^\star | s_i^{\text{query}}, D_i^R). \tag{1}$$

To better understand why DPT is effective for ICRL and to help motivate our proposed frameworks in Section 5, we provide detailed insights into the mechanisms and intuitions behind supervised pretraining. Due to space constraints, see Appendix C.

**Notations.** Let $\sigma(x) = 1/(1 + \exp(-x))$ be the sigmoid function. We use the notation $\succ$ to indicate preference relationships, applicable to both actions and trajectories. For instance, $a \succ a'$ denotes that action $a$ is preferred over $a'$, while $\xi \succ \xi'$ indicates that trajectory $\xi$ is preferred over $\xi'$.

## 4 IN-CONTEXT PREFERENCE-BASED REINFORCEMENT LEARNING

We study two paradigms of preference-based learning: **I-PRL**, which uses step-wise preference labels, and **T-PRL**, which uses trajectory-level preference labels.

---

[1]We assume the test task distribution matches the training one.

## 4.1 PROBLEM SETTINGS

**Overview.** I-PRL and T-PRL primarily differ in the *granularity* of their preference signals and the mechanisms by which preference feedback is generated. From an in-context learning perspective, the key distinction lies in the structure of their context datasets: I-PRL and T-PRL assume *different context dataset formats*, denoted as $D^I$ and $D^T$, respectively. The I-PRL setting generalizes the classical dueling bandit framework (Bengs et al., 2021): at each time step, the agent selects a pair of actions for the current state, receives an *immediate* preference label indicating which action is preferred, and transitions to the next state accordingly. In contrast, the T-PRL setting corresponds to the conventional PbRL paradigm, where feedback is provided over *entire trajectories*, without intermediate or per-step supervision. By jointly investigating I-PRL and T-PRL, we unify and better understand the strengths and challenges of preference-based learning under both fine-grained and long-horizon feedback.

**Latent Reward Functions.** For each task $\tau$, we assume the existence of a latent reward function $R_\tau(s, a)$ that governs preference feedback but is not directly observable by the agent. We also evaluate the performance of a given policy based on the cumulative rewards it achieves under $R_\tau$.

We next describe the settings of I-PRL and T-PRL in more details.

**I-PRL Context.** In I-PRL, for a given task $\tau$, the context dataset $D^I$ is defined as

$$D^I = \{s_1, (a_1, a_1', y_1), s_2, (a_2, a_2', y_2), \ldots, s_H, (a_H, a_H', y_H)\}, \tag{2}$$

where $s_1 \sim \rho_\tau$; for $h \in [H]$, $a_h \sim \pi_{\tau,b}(s_h)$ and $a_h' \sim \pi_{\tau,b}'(s_h)$ are selected by two behavioral policies for task $\tau$, and $y_h \in \{0, 1\}$ is the preference label such that $y_h = 1$ means $a_h$ is more preferred to $a_h'$, denoted as $a_h \succ a_h'$, and vice versa. We follow the convention of PbRL to let the preference $y_h$ follow the Bradley-Terry (BT) model (Bradley & Terry, 1952). Specifically, recall that $\pi_\tau^\star$ is the optimal policy $\tau$ under reward $R_\tau$. Let $A_\tau^\star(s, a) = Q_\tau^\star(s, a) - V_\tau^\star(s)$ be the optimal advantage function for task $\tau$ under the latent reward function $R_\tau(s, a)$, where $Q_\tau^\star(s, a) = \mathbb{E}[\sum_{h=1}^H r_h | s_1 = s, a_1 = a, \pi_\tau^\star]$ and $V_\tau^\star(s) = \mathbb{E}[\sum_{h=1}^H r_h | s_1 = s, \pi_\tau^\star]$. BT model assumes that

$$\mathbb{P}(y_h = 1 | a_h, a_h', s_t, \tau) = \mathbb{P}(a_h \succ a_h' | s_t, \tau) = \sigma\left(A_\tau^\star(s_t, a_h) - A_\tau^\star(s_t, a_h')\right), \tag{3}$$

that is, an action $a$ with higher advantage value (less regret) is more preferred for state $s$ (Hejna et al., 2023; Knox et al., 2022). Lastly, we assume that the state transition follows the preferred action, i.e., $s_{h+1} \sim y_h P_\tau(s_{h+1} | s_h, a_h) + (1 - y_h) P_\tau(s_{h+1} | s_h, a_h')$.

**T-PRL Context.** In T-PRL, preference feedback is provided over pairs of entire trajectories, rather than over individual action choices at specific states. For a given task $\tau$, the context dataset for T-PRL is defined as $D^T = \{\xi, \xi', y\}$, where $\xi = \{s_1, a_1, s_2, a_2, \ldots, s_H, a_H\}$ and $\xi' = \{s_1', a_1', s_2', a_2', \ldots, s_H', a_H'\}$ are two state-action trajectories of horizon $H$, collected by behavioral policies $\pi_{\tau,b}$ and $\pi_{\tau,b}'$, respectively. These trajectories do not contain explicit reward annotations. Instead, we are given a preference label $y \in \{0, 1\}$ that indicates which trajectory is preferred: $y = 1$ signifies that $\xi \succ \xi'$; conversely, $y = 0$ implies that $\xi'$ is preferred. Following standard assumptions in PbRL (Christiano et al., 2017), we also model the preference label $y$ using the BT model:

$$\mathbb{P}(y = 1 | \xi, \xi', \tau) = \mathbb{P}(\xi \succ \xi' | \tau) = \sigma\left(\sum_{h=1}^H R_\tau(s_h, a_h) - \sum_{h=1}^H R_\tau(s_h', a_h')\right). \tag{4}$$

In words, the probability that a trajectory is preferred increases with its cumulative reward.

**Pretraining Datasets.** In both I-PRL and T-PRL settings, we assume a set of pretraining tasks $\{\tau_i\}_{i=1}^m$, each independently sampled from $p_\tau$. For each pretraining task $\tau_i$, we assume a context dataset $D_i$ sampled from $\tau_i$ for pretraining. Depending on the setting (I-PRL or T-PRL), $D_i$ can either follow the structure of $D^I$ or $D^T$, defined as above.

## 4.2 PRACTICALITY OF ICPRL

We first highlight that ICPRL is practical under the same overall preference-label budget commonly used in PbRL, with the added benefit of generalization to new tasks. Moreover, with the advent of powerful large language models (LLMs) and vision langauge models (VLMs), we can considerably reduce the preference-label cost. Lastly, we discuss the value of I-PRL.

**Reallocating the preference budget.** In conventional PbRL, each task consumes many pairwise comparisons to learn a single policy (Christiano et al., 2017). ICPRL redistributes the same total budget across a diverse set of pretraining tasks, then requires only a handful of comparisons at deployment for a new task. Importantly, for the same total annotation effort, ICPRL yields policies that adapt to *unseen* tasks with a few additional labels. This brings considerable practical value.

**Low-cost Annotators.** Recent work shows that LLMs and VLMs can act as preference annotators when prompted with task goals, enabling rapid and inexpensive collection of trajectory-level and even step-wise preferences (Klissarov et al., 2023; Lee et al., 2023). These sources reduce human labeling load and make ICPRL more accessible in practice. Meanwhile, reliability of labels can be guaranteed with LLM techniques such as consensus and calibration. While this is orthogonal to this work, we conduct a pilot study to test the ability of LLM to label trajectory comparisons for ICRL benchmarks, verifying that state-of-the-art LLMs can indeed identify preferred trajectories. See Appendix M.

**Practical value of I-PRL.** Although step-wise labels may appear costly, I-PRL is valuable in settings where instantaneous choices can be judged automatically or cheaply. In particular, it subsumes the dueling bandit problem (Yue et al., 2012), a problem with substantial practical value in many application areas. To this end, I-PRL enables efficient solutions to unseen bandit instances, as demonstrated in our experiments on dueling bandit in Appendix K.1 (space constraint). Moreover, step-wise feedback is information-dense: it improves credit assignment and typically reduces the number of trajectories needed for effective learning, which matters when trajectories are scarce. Lastly, as the first work to propose ICRL with only preference data, we include both T-PRL and I-PRL settings to make the ICPRL framework *complete* and more widely applicable. Our goal is to enable preference-based ICRL in a range of feedback regimes, from weak trajectory comparisons to richer, structured feedback.

## 5 SUPERVISED PRETRAINING FOR ICPRL

We take the first step toward addressing the ICPRL challenge by proposing learning frameworks for both I-PRL and T-PRL. We begin with a straightforward extension of the DPT framework to the reward-free setting, dubbed as *Decision Preference Pretrained Transformer* ($\text{DP}^2\text{T}$). In DPT, each pretraining task $\tau_i$ is associated with a context dataset $D_i^R$ composed of reward-annotated trajectories. In contrast, $\text{DP}^2\text{T}$ utilizes context datasets $D$ constructed solely from preference data. TM policies can be pretrained with $\text{DP}^2\text{T}$ for both I-PRL and T-PRL settings, provided that the context dataset $D$ follows the appropriate format: $D^I$ for I-PRL, with per-step action preferences, and $D^T$ for T-PRL, with trajectory-level comparisons.

As a supervised pretraining framework, $\text{DP}^2\text{T}$ assumes for each pretraining task $\tau_i$, in addition to the context dataset $D_i$ (either $D_i^I$ for I-PRL or $D_i^T$ for T-PRL), there is a query state $s_i^{\text{qe}}$ and an optimal action label $a_i^\star \sim \pi_{\tau_i}^\star(s_i^{\text{qe}})$, sampled from the optimal policy $\pi_{\tau_i}^\star$. Like DPT, $\text{DP}^2\text{T}$ aims to learn a transformer policy $T_\theta(a|s, D)$ capable of inferring the task from context dataset $D$. The pretraining objective remains the same as in (1), where the model is trained to predict $a_i^\star$ given $s_i^{\text{qe}}$ and the context dataset $D_i$. However, unlike DPT which benefits from explicit reward signals in the context dataset, $\text{DP}^2\text{T}$ must infer task-relevant reward information solely from the preference labels in the context dataset. This makes the learning problem more challenging, as the model must learn to interpret implicit reward structure through pairwise comparisons rather than direct supervision. Despite of these challenges, we prove with extensive experiments in Section 7 that $\text{DP}^2\text{T}$ indeed can generalize to new tasks in both of the I-PRL and T-PRL settings.

## 6 PRETRAINING ALGORITHMS BEYOND SUPERVISED PRETRAINING

While $\text{DP}^2\text{T}$ demonstrates strong generalization performance in our experiments, we observe that it does not fully exploit the structure of the available preference data. To address this limitation, we introduce two alternative frameworks, one designed for I-PRL and the other for T-PRL, that leverage preference feedback more directly and effectively. These approaches eliminate the need for *both* high-quality action supervision and reward feedback during pretraining, enabling the agent to pretrain without action labels and generalize to new tasks from preference signals alone. Full mathematical derivations for the results in this section are provided in Appendix I.

**Technical Challenges.** Our frameworks for I-PRL and T-PRL follow a similar principle to standard preference-based reinforcement learning (PbRL): *first estimate the unknown reward function using preference data, then optimize a policy with respect to the learned reward.* However, a key distinction is that PbRL typically focuses on a *single* task, assuming the *same* environment during training and deployment. In contrast, ICPRL needs to estimate a large number of unknown reward functions, one for each pretraining task, which significantly increases the complexity of the problem. This challenge is further compounded in practical settings where only a limited number of trajectories are available per pretraining task, making accurate reward estimation particularly difficult.

**Overview of Frameworks.** Our key insight to address these challenges is, in the same spirit as ICRL, to learn a TM-based *in-context advantage/reward estimator* that interpolates across pretraining tasks and potentially generalize in context to new tasks. Notably, for the I-PRL setting, we propose *In-Context Preference Optimization* (**ICPO**), a framework leveraging a closed-form solution that relates preferences directly to policy optimization. As a result, we can directly optimize TM policies from the preference data without explicitly estimating the task-specific advantage functions. For the T-PRL setting, we propose *In-Context Reward Generation* (**ICRG**) that first learns an in-context reward estimator and uses the learned estimator to label rewards for all the state-action pairs in the pretraining data. Subsequently, we can use any ICRL methods whose pretraining data requirements are satisfied under this scenario to have a TM policy that generalizes to new tasks. In particular, the learned reward estimator is also used during deployment to equip the deployment context datasets with reward information for in-context generalization. We next present our frameworks in details.

## 6.1 PRETRAINING FOR I-PRL

Our goal is a task-conditioned meta-policy $\pi(a|s;\tau)$ that can generalize and solve new tasks. To this end, similar to ICRL, we pretrain a TM policy $T_\theta(a|s, D^I)$. As a meta-policy, $T_\theta(a|s, D^I)$ infers the task information about $\tau$ from the context dataset $D^I$ and takes action accordingly.

To motivate our framework, we begin by observing that for any task $\tau$, if its advantage function $A_\tau^\star$ is known, one can estimate its optimal policy $\pi_\tau^\star(a|s)$ by solving the following regularized objective:

$$\widehat{\pi}_\tau(a|s) \in \operatorname*{argmax}_\pi \mathbb{E}_{s\sim D}\left\{\mathbb{E}_{a\sim\pi(a|s)}\left[A_\tau^\star(a, s)\right] - \beta\operatorname{KL}\left(\pi_\tau^b(\cdot|s)\|\pi(\cdot|s)\right)\right\},$$

where the outer expectation is over states $s$ in the context dataset $D$, and $\pi_\tau^b$ denotes a reference policy for task $\tau$, which serves as a prior for policy learning.

**In-Context Advantage Estimation and Optimization.** In this light, a natural solution is to estimate the advantage functions $\{A_{\tau_i}^\star(s, a)\}_{i=1}^m$ for each pretraining task using a corresponding set of estimators $\{\widehat{A}_{\tau_i}(s, a)\}_{i=1}^m$, *one for each task*. However, this poses a challenging learning problem. Each pretraining task may have only a limited number of trajectories, and this approach fails to exploit any *shared* structure across tasks that could aid interpolation and generalization. To this end, we propose to use an **in-context advantage estimator**, which is a transformer $A_\phi(s, a|D^I)$ that interpolates preference labels *across* pretraining tasks for improved advantage estimation. Specifically, $A_\phi(s, a|D_i^I)$ conditions on the context dataset $D_i^I$ for pretraining task $\tau_i$ to estimate $A_{\tau_i}^\star(s, a)$.

Motivated by the preference modeling assumption of I-PRL in (3), one can train $A_\phi(s, a|D^I)$ with the following maximum likelihood estimation (MLE) objective on the pretraining datasets $\{D_i^I\}_{i=1}^m$:

$$\operatorname*{argmax}_\phi \mathcal{L}^I(\phi) := \frac{1}{m}\sum_{i=1}^m\sum_{h=1}^H y_{i,h}\cdot\sigma\left(A_\phi(s_{i,h}, a_{i,h}|D_i^I) - A_\phi(s_{i,h}, a_{i,h}'|D_i^I)\right)$$
$$+ (1 - y_{i,h})\cdot\sigma\left(A_\phi(s_{i,h}, a_{i,h}'|D_i^I) - A_\phi(s_{i,h}, a_{i,h}|D_i^I)\right),$$

(5)

where $(s_{i,h}, a_{i,h}, a_{i,h}', y_{i,h})$ is the transition tuple at step $h$ in the pretraining dataset of $\tau_i$. After training $A_\phi(s, a|D^I)$, we can pretrain the TM policy $T_\theta(a|s, D^I)$ by

$$\theta \in \operatorname*{argmax}_\theta \sum_{i=1}^m\sum_{h=1}^H \mathbb{E}_{a\sim T_\theta(a|s_{i,h}, D_i^I)}\left[A_\phi(s_{i,h}, a|D_i^I)\right] - \beta\operatorname{KL}[\pi_{\tau_i}^b(\cdot|s_{i,h})\|T_\theta(\cdot|s_{i,h}, D_i^I)].$$

(6)

**In-Context Preference Optimization.** The above two-step procedure, which first learns $A_\phi(s, a|D^I)$ and then solves (6), is a complicated procedure. To address this, motivated by the success of direct

preference optimization methods for RLHF (Rafailov et al., 2023), we observe that the optimization problem in (6) admits a closed-form solution:

$$T_\theta(a|s, D_i^I) = \frac{\pi_{\tau_i}^b(a|s) \cdot \exp\left(A_\phi(s, a|D_i^I)/\beta\right)}{Z(s, \tau_i)}, \tag{7}$$

where $Z(s, \tau) = \sum_a \pi_{\tau_i}^b(a|s) \exp\left(A_\phi(s, a|D_i^I)/\beta\right)$ is a normalizing constant. In particular, Equation (7) directly relates the policy to the advantage function and motivates a reparameterization of $A_\phi(s, a|D_i^I)$: with some algebraic manipulation on (7), we have

$$A_\phi(s, a|D_i^I) = \beta \left(\log T_\theta(a|s, D_i^I) + \log Z(s, \tau_i) - \log \pi_{\tau_i}^b(a|s)\right).$$

In other words, we can parameterize the in-context advantage estimator with the TM policy $\log T_\theta$, and this can be plugged into the learning objective of $A_\phi(s, a|D_i^I)$ in (5) to have

$$\max_\theta \mathcal{L}^I(\theta) := \frac{1}{MH} \sum_{i=1}^M \sum_{h=1}^H \log \sigma \left(\beta \cdot \left(\frac{\log T_\theta(a_{i,h}^+|s_{i,h}, D_i^I)}{\log \pi_{\tau_i}^b(a_{i,h}^+|s_{i,h})} - \frac{\log T_\theta(a_{i,h}^-|s_{i,h}, D_i^I)}{\log \pi_{\tau_i}^b(a_{i,h}^-|s_{i,h})}\right)\right), \tag{8}$$

where $a_{i,h}^+$ and $a_{i,h}^-$ are respectively the preferred and non-preferred actions within the action pair $(a_{i,h}, a_{i,h}')$, indicated by the preference label $y_{i,h}$. By this approach, we can directly optimize the TM policy $T_\theta(a|s; D^I)$ with a simple supervised learning objective rather than first learning an advantage model and then using RL to solve (6).

To further simplify the optimization problem, we choose the reference policy $\pi_{\tau_i}^b$ to be the uniformly random policy for all pretraining tasks $\tau_i$, with three main motivations: **(i)** it leads to consistently strong performance through our experiments in Section 7; **(ii)** the terms containing $\log \pi_{\tau_i}^b$ in (8) now cancel each other and disappear; **(iii)** a uniformly random policy motivates exploration, which can be beneficial for deployment to new tasks. This leads to our final pretraining objective for I-PRL:

$$\max_\theta \mathcal{L}^I(\theta) := \frac{1}{MH} \sum_{i=1}^M \sum_{h=1}^H \log \sigma \left(\beta \cdot \left(\log T_\theta(a_{i,h}^+|s_{i,h}, D_i^I) - \lambda \cdot \log T_\theta(a_{i,h}^-|s_{i,h}, D_i^I)\right)\right), \tag{9}$$

where $\lambda \in (0, 1)$ is a weighting hyperparameter that moderates the influence of non-preferred actions. The role of $\lambda$ is critical: it encourages the transformer policy $T_\theta$ to focus more on increasing the likelihood of preferred actions rather than aggressively suppressing non-preferred ones. Without this adjustment, the model could trivially satisfy the preference constraint by spreading probability mass across many suboptimal actions, without meaningfully boosting the probability of the truly preferred one. Our framework can also be applied for *continuous-control* tasks. See Appendix G.1 for details.

## 6.2 PRETRAINING FOR T-PRL

Because T-PRL is based on a different preference modeling assumption (Equation (4)) from I-PRL (Equation (3)), the framework we propose for I-PRL, which directly optimizes the TM policy, is not applicable in the T-PRL setting. To this end, we are motivated by the principle of *reducing ICPRL problems to standard ICRL formulations*, enabling the reuse of well-established ICRL methods.

**In-Context Reward Estimation.** Specifically, we propose to first estimate the pretraining task reward functions with an transformer-based **in-context reward estimator** $R_\psi(s, a|D^T)$. In particular, $R_\psi(s, a|D_i^T)$ conditions on the context dataset $D_i^T$ for pretraining task $\tau_i$ and estimates its reward function $R_{\tau_i}(s, a)$. Recall that the a context dataset $D^T = \{\xi, \xi', y\}$ for T-PRL contains a trajectory pair and a preference label indicating which trajectory is preferred. Motivated by T-PRL's preference modeling assumption in (4), we train $R_\psi(s, a|D^T)$ with the following MLE objective:

$$\max_\psi \mathcal{L}_R^T(\psi) := \frac{1}{M} \sum_{i=1}^M \log \sigma \left(\sum_{h=1}^H \widehat{R}_\psi(s_{i,h}^+, a_{i,h}^+|D_i^T) - \sum_{h=1}^H \widehat{R}_\psi(s_{i,h}^-, a_{i,h}^-|D_i^T)\right), \tag{10}$$

where $(s_{i,h}^+, a_{i,h}^+)$ is the state-action pair at step $h$ in the preferred trajectory of $D_i^T$ while $(s_{i,h}^-, a_{i,h}^-)$ is the state-action pair in the non-preferred one.

**Reward Labeling for ICRL.** With a learned reward estimator $R_\psi(s, a|D^T)$, we can label all state-action pairs in the pretraining datasets. Specifically, for each state-action pair $(s_{i,h}, a_{i,h})$

in either the preferred or non-preferred trajectory of $D_i^T$, we append to it an estimated reward $\widehat{r}_{i,h} = R_\psi(s_{i,h}, a_{i,h}|D_i^T)$. By this approach, we now have for each pretraining task $\tau_i$ a pretraining dataset containing two trajectories with reward information. This enables us to leverage *off-the-shelf* ICRL methods to obtain a pretrained TM policy $T_\theta^R(a \mid s, D^R)$ that generalizes to new tasks when conditioned on a context dataset $D^R$ *containing reward information*. Crucially, ICRG uses the learned reward estimator $R_\psi(s, a \mid D^T)$ to infer such reward signals, allowing us to deploy $T_\theta^R(a \mid s, D^R)$ even when the original context datasets contain only preference data. See Algorithm 2 in Appendix N for pseudocode of the complete pipeline. In particular, we choose the *Decision Importance Transformer* framework (Dong et al., 2025), which has the least pretraining data requirements and is specifically designed for pretraining datasets containing suboptimal trajectories. Due to its tangential connection to our contribution, we introduce and discuss it in detail in Appendix J.

**Framework Comparisons and Policy Deployments.** To help understand all the proposed ICPRL frameworks, we provide a summary of them with their corresponding models. In addition, we elaborate on their deployments. Due to space constraint, we defer these discussions to Appendix H.

# 7 EXPERIMENTS

We empirically demonstrate the efficacy of our proposed frameworks through experiments on various dueling bandit and MDP problems. Due to space constraints, we defer all the bandit experiments to Appendix K.1 and mainly present the more challenging MDP experiments. See Appendix G for implementation details and visualizations of model architectures.

**Experiments Setup.** We conduct experiments for both I-PRL and T-PRL settings. We consider two representative ICRL tasks: the challenging navigation task **DarkRoom** (Laskin et al., 2022) and the complex continuous control task **Meta-World** (reach-v2) (Yu et al., 2020). In DarkRoom, the agent is randomly placed in a room with discrete grids. The goal of agent is to move to an *unknown* goal location on one of the grids. The agent has $5$ actions and needs to reach the goal in $H = 100$ steps. In Meta-World, the agent controls a robotic hand to reach a target position in 3D space. We highlight that we use different tasks for pretraining and deployment so that all the test tasks are *unseen* to the ICPRL models. See Appendix D for more details.

**Pretraining Datasets.** To collect pretraining datasets for DarkRoom, we use two behavioral policies: the uniformly random policy and a policy that at every step, with probability $p$ (respectively $1 - p$) follows the optimal policy (respectively the uniformly random policy) to choose action. For Meta-World, we construct the pretraining datasets using training checkpoints of *Soft Actor Critic* (SAC). Specifically, SAC is trained until convergence for each task, and we use checkpoints with $\%30$ and $\%80$ performance of the optimal policy as the behavioral policies. Due to their different structure, I-PRL and T-PRL follow different data generation processes. See Appendix F for details.

**Performance Metric.** We evaluate policies using the standard trajectory cumulative reward. During deployment to a task $\tau$, we use the ground-truth reward function $R_\tau(s, a)$, which remains unobservable to the agent, to assign true rewards to all state-action pairs. The total reward is computed by summing these values over the rollout horizon.

**Baselines.** To benchmark the performance of our proposed frameworks, we compare against two strong baselines: the state-of-the-art ICRL method **DPT**; and **SAC**, a widely used online RL algorithm that trains an agent from scratch in each environment. Implementation details for both baselines are provided in Appendix E. Importantly, comparison to DPT is *inherently unfair*, as it relies on *full reward supervision* and *costly optimal action labels* during pretraining. We include DPT as an *oracle* baseline to contextualize the performance of preference-based methods. To further benchmark the difficulty of the test tasks, we report the performance of SAC trained from scratch for $1000$ episodes and select the *best* checkpoint for evaluation. In contrast, ICPRL models are deployed to new tasks using at most three offline preference trajectories, underscoring their practicality.

**Results for I-PRL.** We first observe that both of our ICPRL methods almost always *significantly improve over the behavioral policies* of the context datasets, demonstrating their generalization capabilities. For **DarkRoom**, DPT consistently demonstrates the best performance, This is not surprising, as it uses considerably more information for both pretraining and testing. However, when the context dataset has **high-quality**, our ICPRL methods pretrained and deployed *without*

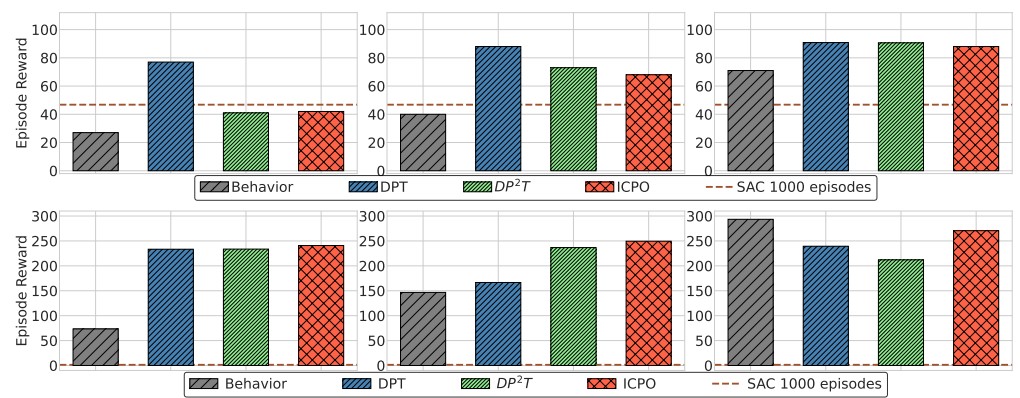

Figure 2: I-PRL results under context datasets of varying quality (left to right: **low**, **medium**, and **high** quality) in DarkRoom (top) and Meta-World.

any reward information can match DPT. Importantly, ICPO, without using any optimal action labels for pretraining, demonstrates comparable performance to $DP^2T$ in all cases, showing its *effective usage* of the preference pretraining data. Quite surprisingly, for **Meta-World**, ICPO *consistently outperforms* two supervised pretraining methods, even DPT with full reward supervision. We attribute this to the fact that in more complex, continuous domains, action labels are harder to interpret and less informative. As a result, ICPO, which optimizes directly from preference feedback without relying on reward or action labels, demonstrates superior adaptability under these conditions.

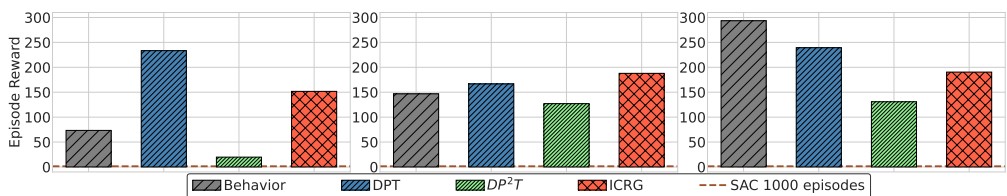

Figure 3: Meta-World (T-PRL) results with context datasets of **low**, **medium**, and **high** quality.

**Results for T-PRL.** Due to space constraints, we present the results for the more challenging Meta-World (T-PRL) problem and defer results for DarkRoom (T-PRL) to Appendix K. We first note that, due to sparse (trajectory-level) preference labels, T-PRL is considerably more challenging than I-PRL with step preference. As a result, ICPRL models (ICRG and $DP^2T$) pretrained in T-PRL setting demonstrate less competitive performance than their counterparts in the I-PRL setting. However, for both **low** and **medium** context datasets, our method ICRG still significantly improves over the behavioral policies. In addition, ICRG consistently outperforms $DP^2T$, showcasing it effectively recovering reward-relevant information from trajectory-level preferences. Most importantly, our ICPRL methods considerably outperform online learning from scratch in the test tasks (SAC with 1000 episodes), demonstrating their promises towards competitive in-context decision-makers with *completely reward-free pretraining and deployment*.

## 8 DISCUSSION AND LIMITATIONS

We introduced ICPRL, a new paradigm for training in-context transformer policies without explicit rewards, using only preference supervision. We proposed frameworks for both I-PRL (per-step) and T-PRL (trajectory-level) settings, and showed that pretrained policies generalize effectively across tasks. A key limitation is the assumption of pre-collected offline preference data; future work could explore active preference querying during pretraining to improve data efficiency and generalization.

ETHICS STATEMENT

This study complies with the ICLR Code of Ethics. All datasets employed are publicly available and open-source under licenses that permit research use. No private or personally identifiable information was accessed, and no new data were collected from human subjects. The research does not pose privacy, security, or fairness concerns. The authors declare no conflicts of interest and no external sponsorship.

REPRODUCIBILITY STATEMENT

All datasets used in our experiments are publicly available and described in Section 7 and Appendix D. Implementation details of baselines, model architecture and hyper-parameter settings are provided in Appendix G. We also attach our codebase as Supplementary Material. Complete derivations of theoretical results and assumptions underlying our analysis are included in Appendix I. These resources together allow independent researchers to verify and reproduce our findings.

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

## A  USE OF LARGE LANGUAGE MODELS (LLMS)

Large Language Models (LLMs) were used solely as general-purpose assistive tools to improve the clarity and readability of the manuscript. Specifically, we used an LLM to help rephrase and polish text that we had already drafted.

## B  RELATED WORK

**Preference-based Reinforcement Learning (PbRL).** PbRL focuses on learning from preference signals, typically given as comparisons between actions or trajectories, instead of relying on scalar reward functions (Christiano et al., 2017; Wirth et al., 2017; Brown et al., 2019; Fürnkranz et al., 2012). Notably, the well-known dueling bandit problem Yue et al. (2012); Wu & Liu (2016); Dudík et al. (2015); Komiyama et al. (2015) is a special case of the PbRL problem without state transitions. A common paradigm in PbRL involves a two-stage model-based process: first, learning a reward function from preferences, and second, optimizing a policy using standard RL algorithms with the learned reward (Ibarz et al., 2018; Lee et al., 2021; Liu et al., 2023; 2022). The most salient distinction between the proposed ICPRL paradigm and prior PbRL approaches is that PbRL assumes the *same* environment for training and evaluation, whereas ICPRL operates in the in-context setting, where a model must generalize to *new* tasks at inference time without parameter updates.

**Offline Reinforcement Learning.**  Our work is also related to offline RL(Levine et al., 2020; Matsushima et al., 2020; Prudencio et al., 2023). The goal of offline RL is to learn an optimal policy from pre-collected datasets without exploration and interacting with the environments (Wu et al., 2019; Kidambi et al., 2020; Kumar et al., 2020; Rashidinejad et al., 2021; Yin & Wang, 2021; Jin et al., 2021; Dong et al., 2023; Fujimoto & Gu, 2021). In particular, offline RL methods assume that the training data is collected from the environment to be deployed for. To this end, our work falls into the ICRL paradigm, and the goal is to learn a meta-policy from offline preference data collected from diverse RL tasks to generate to unseen RL tasks.

**Transformer Models and In-Context Reinforcement Learning.** Transformer models have demonstrated superior performance in RL problem (Li et al., 2023; Yuan et al., 2023). Building on the in-context learning abilities of transformers, ICRL methods aim to learn a TM-based meta-policy to generalize to unseen tasks in context. ICRL methods differ in their requirements of context datasets. For example, *Algorithm Distillation* (**AD**) (Laskin et al., 2022) and its variations (Zisman et al., 2023; Tarasov et al., 2025) uses sequential modeling to emulate the learning process of RL algorithms, i.e., meta-learning (Vilalta & Drissi, 2002). Decision transformer (Chen et al., 2021)(DT)-based methods rely on return-to-go to guide the transformer models to generalize to new tasks (Grigsby et al., 2023; Huang et al., 2024; Schmied et al., 2024). A recent framework *Decision Pretrained Transformer* (**DPT**) uses supervised-pretraining for in-context decision making. DPT trains transformers to predict the optimal action given a query state and a set of transitions. Despite its strong performance, DPT requires high-quality action labels for its pretraining. However, to our best knowledge, all existing ICRL methods assume explicit reward (goal) signal for pretraining. To this end, we make the first step towards reward/goal-free ICRL.

## C  WHY SUPERVISED-PRETRAINING WORKS FOR ICRL

As rigorously proved in Lee et al. (2024), the supervised pretraining approach for ICRL can be interpreted as training the TM policy $T_\theta$ to conduct an implicit Posterior Sampling (PS) Osband et al. (2013). Under this perspective, during deployment, the decision-making process of a pretrained TM $T_\theta$ is equivalent to conducting three consecutive moves:

(i) When given the context dataset $D$, $T_\theta$ first *implicitly* constructs a posterior distribution $p(\tau|D)$ over tasks;

(ii) $T_\theta$ samples a task $\tau'$ following the constructed posterior $p(\tau|D)$;

(iii) $T_\theta$ follows the optimal policy of the sampled task $\tau'$, i.e., $T_\theta(a|s, D) \approx \pi^\star_{\tau'}(s)$.

In particular, similar to other Bayesian approaches, as the size of context dataset $|D|$ increases, the implicit posterior $p(\tau|D)$ concentrates toward the true test task $\tau^{\text{test}}$, and the sampled task $\tau'$ becomes

more similar to $\tau^{\text{test}}$. As a result, if the pretrained TM $T_\theta$ follows the optimal policy for $\tau'$, its performance also increases in $\tau^{\text{test}}$.

# D  MDP ENVIRONMENT DETAILS

We evaluate our proposed methods on two sequential decision-making environments with contrasting characteristics—one discrete with sparse-reward (DarkRoom) and one continuous with dense-reward (Meta-World Reach-v2)—to assess generalization across task modalities.

**DarkRoom.**  DarkRoom is a grid-based navigation task with sparse rewards, originally proposed by Laskin et al. (2022). The environment is a $10 \times 10$ grid, where each cell represents a discrete location the agent can occupy. At the start of each episode, the agent is placed at a random location, and the goal location, unknown to the agent, is randomly chosen and then fixed throughout the episode. The agent can take one of five discrete actions at each time step: up, down, left, right, or no-op. The episode ends either after 100 time steps (the maximum horizon) or upon reaching the goal. A reward of 1 is given only when the agent steps into the goal location; otherwise, all rewards are 0.

We pretrain on 80 randomly sampled tasks (each with a different goal position) and evaluate on a held-out set of 20 tasks. For each task, we generate preference-labeled trajectories using behavior policies that mix optimal and random behavior (see Appendix F). The discrete action space and binary reward make DarkRoom a useful testbed for evaluating how well our models can leverage sparse preference signals to infer task objectives.

**Meta-World (Reach-v2).**  Reach-v2 is a continuous control task from the ML1 benchmark of the Meta-World suite (Yu et al., 2020). It requires controlling a 7-DoF Sawyer robotic arm to move its end-effector to a 3D target position, which is randomly sampled at the beginning of each task instance. The agent observes both its proprioceptive state (joint positions and velocities) and the target position. The action space is continuous and represents end-effector displacements. At each time step, the agent receives a reward equal to the negative Euclidean distance to the goal. The episode length is fixed at a horizon of 150 time steps.

We use 45 tasks for pretraining and hold out 5 tasks for evaluation. Each task is defined by a different target goal sampled from a bounded region in 3D space. To generate pretraining datasets, we train SAC policies to convergence for each task, and select intermediate checkpoints to simulate suboptimal behavior. Preference labels are generated by comparing trajectories collected from these policies. Due to the continuous nature of the action space and the dense reward function, Reach-v2 is a significantly more challenging setting compared to DarkRoom, providing a strong test of the scalability and adaptability of preference-based models.

**Deployment.** Deployments for both environments are under the offline RL setting: during deployment, no environment interaction is permitted beyond the given offline context. Agents must rely entirely on the offline, preference-labeled context data to infer the task and generalize their behavior.

# E  BASELINE IMPLEMENTATION DETAILS

To benchmark the performance of our proposed ICPRL methods in both the I-PRL and T-PRL settings, we compare against several strong baselines. Each of these baselines represents a distinct paradigm in reinforcement learning: fully supervised transformer-based meta-RL, hybrid preference-supervised methods, and classical model-free reinforcement learning from scratch.

**Decision-Pretrained Transformer (DPT).**  DPT (Lee et al., 2024) is a transformer-based meta-RL method that performs in-context adaptation through supervised pretraining. The model learns to predict optimal actions by conditioning on a query state and a reward-labeled trajectory drawn from the same task. During pretraining, DPT is provided with full reward supervision and oracle optimal action labels. For each training task, we use a converged SAC policy to generate reward-annotated context trajectories and extract optimal actions for randomly sampled query states. The model is trained to maximize the log-likelihood of these optimal actions given the context and the query state.

At test time, the pretrained transformer is deployed in a reward-rich offline setting, where it receives reward-annotated context trajectories from unseen tasks and is queried at each step to produce an action. As DPT represents the strongest fully supervised baseline (with access to reward and optimal labels), it serves as an upper bound in our comparison.

**DP$^2$T (DPT with Preference-style Context).** This baseline adapts the DPT framework to our reward-free setup by changing the structure of the context data. Instead of using reward-annotated transitions, it consumes preference-labeled trajectories generated from the I-PRL pipeline. However, it retains access to optimal actions for query states during pretraining. In this sense, DP$^2$T is a hybrid method—it leverages the structure of our proposed setting while retaining strong supervision through optimal action labels.

The goal of this baseline is to isolate the effect of removing reward supervision in the context while still keeping access to action supervision. By comparing it against DPT (with full reward and action supervision), and ICPO and ICRG (with neither reward nor action supervision), we can disentangle the individual contributions of reward-rich context and label-rich supervision.

**Soft Actor-Critic (SAC).** SAC (Haarnoja et al., 2018) is a widely used model-free deep RL algorithm that learns stochastic policies in continuous or discrete environments by maximizing a trade-off between expected return and policy entropy. It serves as a strong baseline for traditional reinforcement learning that does not leverage offline in-context data.

In our experiments, SAC plays two roles. First, we use it to train behavioral policies for generating the pretraining datasets in both I-PRL and T-PRL settings. For each pretraining task, we train SAC to convergence and then choose intermediate checkpoints to simulate suboptimal policies (e.g., checkpoints with 20%, 40%, and 80% of the return of the final converged policy) for context trajectory generation. This allows us to construct preference-labeled data for ICPRL training.

Second, we use SAC as a standalone baseline to measure how well a task-specific policy can perform when trained from scratch. For each test task, we train a SAC agent independently using 1,000 episodes of environment interaction. We implement SAC using Stable Baselines3 (Raffin et al., 2021) with default hyperparameters and a fixed episode horizon (100 for DarkRoom, 150 for Meta-World Reach-v2).

Together, these baselines represent a diverse set of RL paradigms and help us contextualize the benefits and trade-offs of purely preference-based pretraining for transformers to generalize to new tasks.

# F    PRETRAINING DATA GENERATION

We construct our pretraining datasets to simulate realistic, reward-free supervision via preferences. Our approach leverages varying-quality behavioral policies to provide diverse experiences across tasks. We separately describe the data generation process for each environment and for both the I-PRL and T-PRL settings.

**DarkRoom.** To construct the context dataset, we generate trajectories using a **mixed policy** that interpolates between the optimal and random policies. Specifically, at every time step, the action is chosen following the optimal policy with probability $p$, and with probability $1 - p$ following a uniformly random policy. We adjust $p$ to control the overall trajectory return, ensuring the resulting policies yield approximately 20%, 40%, or 80% of the optimal cumulative reward. This allows us to evaluate how the quality of context dataset impacts generalization.

For the **I-PRL** setting, at each time step $h$, we sample a pair of actions, one from the uniformly random policy and the other from a mixed policy defined as above. Then we compute the preference labels using the (optimal) advantage function. We note that the optimal advantage function for DarkRoom has a closed form. After the preference label is generated, the current state transits according to the preferred action. We collect 1000 trajectories for each task.

In the **T-PRL** setting, to construct a pair of trajectories, we rollout twice a mixed policy defined above to have two trajectories. Then the trajectory preference labels are generated following the BT

model defined in Equation (4). We note that the true reward functions are only for preference label generation, and they are not accessible during pretraining. For each task, we repeat this process to have 5000 trajectory pairs and their associated preference labels.

**Meta-World (Reach-v2).** Each task in the Meta-World ML1 Reach benchmark is defined by a unique 3D goal position for the robotic arm. We train a separate SAC agent per task to convergence, then select multiple checkpoints to serve as behavioral policies with different proficiency levels. Specifically, we extract checkpoints that achieve approximately 20%, 40%, and 80% of the final SAC return and use them to generate demonstration trajectories. These trajectories vary in quality and coverage, offering a broad distribution of behaviors for preference comparisons.

In the **I-PRL** setup, at each time step $h$, we sample a pair of actions, one from the checkpoint with 20% of the final return and the other from the checkpoint with 80% of the final return. Then we compute the preference labels using the (optimal) advantage function. Since the optimal advantage function for Meta-World is unknown, we approximate it using the value functions learned by converged SAC policies. Let $\widehat{Q}(s, a)$ be the learned action-value function of a converge SAC policy. We approximate the optimal advantage function with $A^\star(s, a) \approx \widehat{Q}(s, a) - \mathbb{E}_{a \sim \pi}[\widehat{Q}(s, a)]$ where $\pi$ is the converged SAC policy and we estimate the expectation with Monte Carlo simulation, i.e., sampling a lot of actions from $\pi$ and average their $\widehat{Q}(s, a)$ values. After the preference label is generated, the current state transits according to the preferred action. We collect 1000 trajectories for each task.

In the **T-PRL** setup, we first sample two trajectories with the SAC checkpoints with 50% return of the converged policies. Then the trajectory preference labels are generated following the BT model defined in Equation (4). We note that we only use the true reward functions during preference label generation, these reward functions are unobservable during pretraining. For each task, we repeat this process to have 5000 trajectory pairs and their associated preference labels.

# G  IMPLEMENTATION DETAILS

All transformer models use the GPT-2 architecture (Radford et al., 2019). We adopt causal masking and position embeddings as in standard autoregressive transformers. For all experiments in the I-PRL setting, we use transformers with 6 layers, 256 hidden dimensions, and 8 attention heads. For experiments in the T-PRL setting, we use transformer models with 4 layers, 32 hidden dimensions, and 4 attention heads for DarkRoom and models with 8 layers, 256 hidden dimensions, and 8 attention heads for Meta-World. For both policy models $T_\theta$ and reward estimators $R_\psi$, we format the input sequence as tokenized tuples of states, actions, preferences, next states, and/or rewards depending on the setting.

**Transformer Policy Architecture.** Figure 4 shows the architecture of the TM policy $T_\theta$ for both the T-PRL and I-PRL settings.

**Reward Model Architecture.** For the in-context reward estimator $R_\phi(s, a|D^T)$, we use two separate modules to independently embed the transitions from the preferred and non-preferred trajectories, respectively denoted as $\xi^+$ and $\xi^-$. Specifically, conditioned on the context dataset $(\xi^+, \xi^-)$, $R_\phi(s, a|D^T)$ takes as input a query state-action pair and outputs an estimated reward for the given query state-action pair. Figure 5 illustrates its design and model architecture. When using $R_\phi(s, a|D^T)$ to estimate rewards, either to pretrain the DIT model or during deployment, we read out its prediction at the last time step as the estimated reward.

**Optimization.** All models are trained using the Adam optimizer with a learning rate of $3 \times 10^{-4}$ and batch size of 32. We use early stopping based on validation performance on a small hold-out pretraining dataset.

## G.1  ICPO FOR CONTINUOUS CONTROL TASKS

**Continuous Control.** In the continuous control setting, we model the policy as a Gaussian distribution $\mathcal{N}(a; T_\theta(s, D^I), \gamma \mathbf{I})$, where the transformer-based policy $T_\theta$ predicts the mean and $\gamma$ controls the

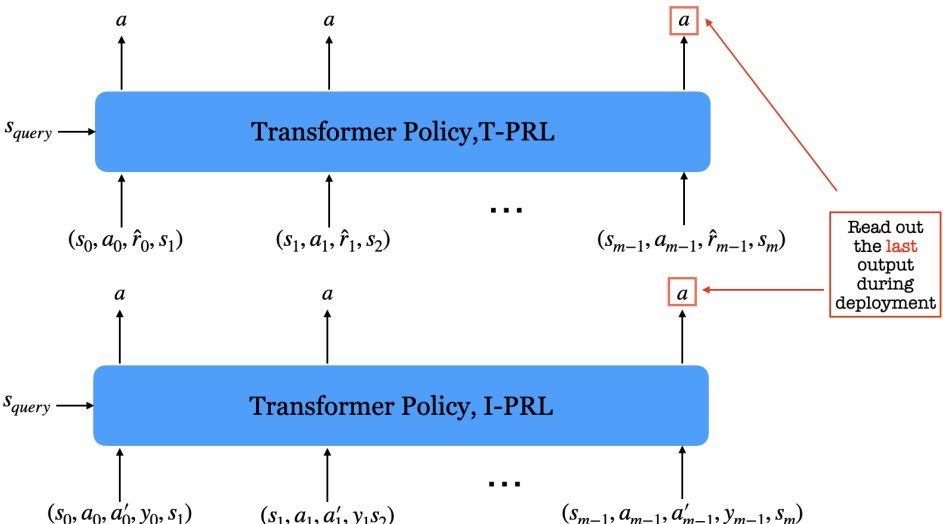

Figure 4: Transformer policy architecture. The **bottom** model illustrates the architecture of TM policies for the I-PRL setting. The **top** depicts the T-PRL setting where the reward values $\widehat{r}_h$ are approximated by the in-context reward estimator.

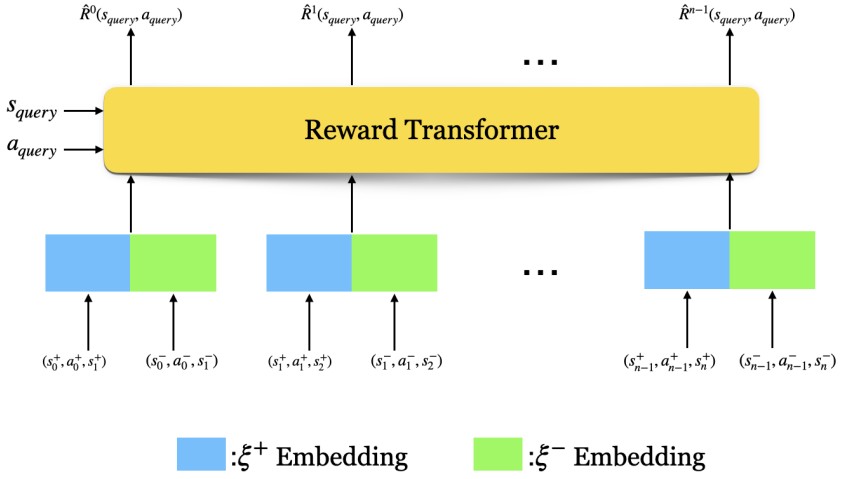

Figure 5: The architecture of the reward transformer.

fixed diagonal covariance. Under this formulation, the pretraining objective simplifies to:

$$\frac{1}{MH}\sum_{i=1}^{M}\sum_{h=1}^{H}\log\left(\sigma\left(\beta\cdot\left(\|a_{i,h}^{-}-T_\theta(s_{i,h},D_i^I)\|_2-\|a_{i,h}^{+}-T_\theta(s_{i,h},D_i^I)\|_2\right)\right)\right),\qquad(11)$$

where $a_{i,h}^{+}$ and $a_{i,h}^{-}$ denote the preferred and less-preferred actions at step $h$ in task $\tau_i$. See its full derivation in Section I.

## H    FRAMEWORK COMPARISONS AND POLICY DEPLOYMENTS

We summarize the ICPRL frameworks introduced so far along with their corresponding models: **(a) DP²T** yields task-conditioned policies $T_\theta^S(a|s, D^I)$ for I-PRL and $T_\theta^S(a|s, D^T)$ for T-PRL, conditioning on different context datasets; **(b) ICPO** (I-PRL only) produces a similar policy $T_\theta^I(a \mid s, D^I)$ with the same inference structure as DP²T but differs in pretraining: it does not require

optimal action labels and is trained with a preference-based supervised learning objective. **(c) ICRG** (T-PRL only) outputs an in-context reward estimator $R_\psi(s, a|D^T)$ and an ICRL policy $T_\theta^R(a|s, D^R)$ requiring reward-labeled context data.

**Deployment Setup.** Due to the different structures of I-PRL and T-PRL, these models have distinct deployment procedures. In both settings, we first sample an unseen test task $\tau^{\text{test}} \sim p_\tau$.

**DP$^2$T Deployment.** In both I-PRL and T-PRL, DP$^2$T assumes access to a context dataset $D_{\tau^{\text{test}}}$ collected from an unknown behavior policy in the test task $\tau^{\text{test}}$. This dataset corresponds to $D_{\tau^{\text{test}}}^I$ in I-PRL and $D_{\tau^{\text{test}}}^T$ in T-PRL. At each time step $h$, the agent observes state $s_h$ and samples an action from the TM policy $a_h \sim T_\theta^S(a_h|s_h, D_{\tau^{\text{test}}})$.

**ICPO Deployment (I-PRL).** Similar to DP$^2$T, ICPO assumes access to a context dataset $D_{\tau^{\text{test}}}^I$. At each step $h$, the agent acts according to the TM policy $a_h \sim T_\theta^I(a_h|s_h, D_{\tau^{\text{test}}}^I)$.

**ICRG Deployment (T-PRL).** A context dataset $D_{\tau^{\text{test}}}^T$ is first sampled. Subsequently, a behavior trajectory with only state-action pairs $D = \{s_1, a_1, s_2, a_2, \dots, s_H, a_H\}$ is sampled from $\tau^{\text{test}}$. Using the pretrained in-context reward estimator $R_\psi$, each state-action pair $(s_h, a_h)$ is annotated with a predicted reward $\widehat{r}_h = R_\psi(s_h, a_h|D_{\tau^{\text{test}}}^T)$. The resulting reward-augmented context $D^A = \{s_h, a_h, \widehat{r}_h\}_{h=1}^H$ is used to prompt the ICRL policy $a_h \sim T_\theta^R(a_h|s_h, D^A)$ to take actions $a_h$ at each step $h$.

# I  DERIVATIONS

## I.1  DERIVATION OF EQUATION (7)

For any fixed task $\tau$, state $s$, context dataset $D^I$ and reference policy $\pi_\tau^b$ for $\tau$, we have

$$\max_\pi \mathbb{E}_{a \sim \pi(a|s; D^I)} \left[ A_\phi\left(s, a|D^I\right) - \beta \cdot \text{KL}(\pi(\cdot|s; D^I)\|\pi_\tau^b(\cdot|s)) \right]$$

$$= \min_\pi \mathbb{E}_{a \sim \pi(a|s; D^I)} \left[ \log \frac{\pi(a|s; D^I)}{\pi_\tau^b(a|s)} - \frac{1}{\beta} A_\phi\left(s, a|D^I\right) \right]$$

$$= \min_\pi \mathbb{E}_{a \sim \pi(a|s; D^I)} \left[ \log \frac{\pi(a|s; D^I)}{\pi_\tau^b(a|s) \exp(A_\phi\left(s, a|D^I\right)/\beta)} \right]$$

$$= \min_\pi \mathbb{E}_{a \sim \pi(a|s; D^I)} \left[ \log \frac{\pi(a|s; D^I)}{\pi_\tau^b(a|s) \exp(A_\phi\left(s, a|D^I\right)/\beta)/Z(s, \tau)} - \log Z(s, \tau) \right]$$

$$= \min_\pi \mathbb{E}_{a \sim \pi(a|s; D^I)} \left[ \log \frac{\pi(a|s; D^I)}{\pi_\tau^b(a|s) \exp(A_\phi\left(s, a|D^I\right)/\beta)/Z(s, \tau)} \right] \quad (Z(s, \tau) \text{ is independent of } \pi)$$

$$= \min_\pi \text{KL}(\pi(\cdot|s; \tau)\|\pi_\tau^\star),$$

where $\pi_\tau^\star(a|s) = \pi_\tau^b(a|s) \exp(A_\phi\left(s, a|D^I\right)/\beta)/Z(s, \tau)$. Note that the optimum $\pi$ for a fixed $s$ and task $\tau$ is obtained at $\pi = \pi_\tau^\star$, which is unique by the uniqueness property of KL divergence, i.e., $\text{KL}(\pi\|\pi_\tau^\star) = 0$ if and only if $\pi = \pi_\tau^\star(a|s)$.

## I.2  DERIVATION OF EQUATION (11)

In the case of continuous case, with a notation overload, we choose $T_\theta(a|s; D^I) \sim \mathcal{N}(a; T_\theta(s, D^I), \gamma\mathbf{I})$ such that we use the TM policy to directly predict the mean action $T_\theta(s, D^I) \in \mathbb{R}^d$ where $d$ is the action dimension. Thus, we have

$$\log T_\theta(a|s, D_i) = \log C(d, \gamma) - \frac{1}{2\gamma}\|a - T_\theta(s, D^I)\|_2^2,$$

where $C(d, \gamma)$ is a constant only involving universal constants, $d$ and $\gamma$. With this identity, for any state $s$ and pair of actions $a^+$ and $a^-$,

$$\log \sigma\left(\beta \cdot \left(\log T_\theta(a^+|s, D^I) - \lambda \cdot \log T_\theta(a^-|s, D^I)\right)\right)$$

$$= \log \sigma\left(\beta\left(\log C(d, \gamma) - \frac{1}{2\gamma}\|a^+ - T_\theta(s, D^I)\|_2^2 - \log C(d, \gamma) + \frac{1}{2\gamma}\|a^- - T_\theta(s, D^I)\|_2^2\right)\right)$$

$$= \log \sigma\left(\beta \cdot \left(\|a^- - T_\theta(s, D^I)\|_2^2 - \|a^+ - T_\theta(s, D^I)\|_2^2\right)\right).$$

## J  DECISION IMPORTANCE TRANSFORMER

The *Decision Importance Transformer* (DIT) framework addresses the challenge of obtaining optimal action labels for query states during pretraining (Dong et al., 2025). We use DIT as a module for our proposed framework for the T-PRL setting, specifically for pretraining without query states and optimal action labels. During pretraining, DIT requires for each pretraining task $\tau_i$ a context dataset $D_i = \{\xi_i\}$ containing a trajectory $\xi_i$ with reward information. For simplicity, we assume that each context dataset only contains one trajectory.

To apply DIT in our ICPRL setting, we first pretrain an in-context reward estimator $R(s, a|D^I)$ and then use it to label all the trajectory pairs in the context dataset. After this step, we have trajectories with no missing reward information (all the rewards are estimated by $R(s, a|D^I)$), and we use DIT to pretrain a TM policy that can generalize to new tasks. In particular, we also rely on $R(s, a|D^I)$ to generate all the required reward information during deployment so that the DIT pretrained models, which require reward information in the context dataset, can be deployed in new tasks. Next we discuss DIT in detail.

**Optimization Objective.**  DIT approximates the advantage of a state-action pair by fitting two transformers in parallel. Let $G_h^i = \sum_{h'-h}^{H} \gamma^{h'-h} r_h^i$ denote the in-trajectory discounted cumulative reward for the trajectory $\xi_i$ with reward information in the context dataset $D_i$ for task $\tau_i$. For each state-action pair, $\hat{Q}_\zeta(s_h^i, a_h^i|D_Q^{h,i})$ and $\hat{V}_\psi(s_h^i|D_V^{h,i})$ estimate the action-value and state-value functions, respectively. We compute the discounted cumulative rewards at each step of a trajectory and construct training datasets of the form $D_Q^i = \{(s_h^i, a_h^i, G_h^i)\}_{h=1}^{H-1}$ and $D_V^i = \{(s_h^i, G_h^i)\}_{h=1}^{H-1}$ for every trajectory. The V-value transformer takes a sequence of states as input to estimate state values, while the Q-value transformer processes state-action pairs to approximate Q-values. Figure 6 illustrates the two transformers, each processing a different input sequence. We train $\hat{Q}_\zeta$ and $\hat{Q}_\zeta$ with the following objective function:

$$\min_{\zeta,\theta} \sum_{i=1}^{m} \sum_{h=1}^{H} (\hat{Q}_\zeta(s_h^i, a_h^i|D_Q^{h,i}) - G_h^i)^2 + (\hat{V}_\psi(s_h^i|D_Q^{h,i}) - G_h^i), \qquad (12)$$

where $m$ denotes the total number of trajectories in the pretraining dataset, and $D_Q^{h,i}$ and $D_V^{h,i}$ contains the first $h$ tuples of $D_Q^i$ and $D_V^i$. After training, we calculate the advantage value of a state-action pair for task $\tau_i$ as

$$\hat{A}_b(s_h^i, a_h^i|\tau_i) = \hat{Q}_\zeta(s_h^i, a_h^i|D_Q^{h,i}) - \hat{V}_\phi(s_h^i|D_V^{h,i}). \qquad (13)$$

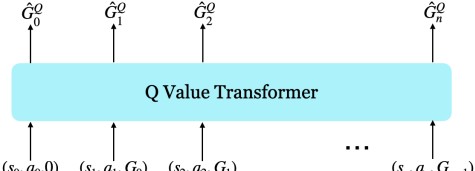
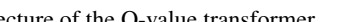
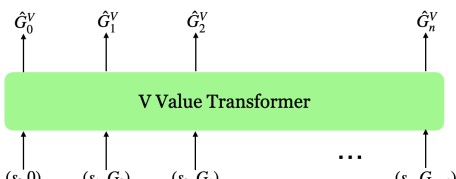

(a) The Architecture of the Q-value transformer.       (b) The Architecture of the V-value transformer.

Figure 6: Overview of the DIT architecture. The transformers approximate the Q and V values separately, which are then used to compute the advantage of a specific state-action pair.

$\hat{A}(s_h^i, a_h^i|\tau^i)$ works as an approximation to the true advantage function, we now could use it for the optimization of the policy transformer $T_\theta^R$ as

$$\theta^\star \in \operatorname*{argmin}_{\theta \in \Theta} -\frac{1}{mH} \sum_{i=1}^{m} \sum_{h=1}^{H} \exp(\hat{A}_b(s_h^i, a_h^i|\tau_i)/\eta) \log T_\theta^R(a_h^i|s_h^i, D^i). \qquad (14)$$

Intuitively, for a trajectory $\tau_i$ from a task, DIT optimizes the TM policy with a weighted supervised pretraining objective where the weights are calculated through the advantage function. In particular, it assigns more weights for better actions, i.e., actions with higher advantage values.

**Implementation of DIT's Value Transformers.** For DarkRoom, we use a GPT-2 architecture with 4 layers, 32 hidden dimensions, and 4 attention heads, as originally reported by (Lee et al., 2024). Meta-World (Reach-v2) requires a larger model, so we adopt a GPT-2 with 8 layers, 256 hidden dimensions, and 8 attention heads. Algorithm 1 provides the pseudocode for training DIT.

---

**Algorithm 1** Pretraining of Decision Importance Transformer

---

1: **Input:** Pretraining Dataset $\mathcal{D} = \{D^i\}$; transformer models $T_\theta, \widehat{Q}_\zeta, \widehat{V}_\phi$.
2: // In-context Estimation of Advantage Functions
3: Randomly initialize and train $\widehat{Q}_\zeta$ and $\widehat{V}_\phi$ by optimizing the loss in Equation (12).
4: Construct the in-context advantage estimator as:

$$\widehat{A}_b = \widehat{Q}_\zeta - \widehat{V}_\phi.$$

5: //  Weighted Pretraining
6: Randomly initialize $T_\theta$.
7: With trained $\widehat{A}_b$ and $\mathcal{D}$, train $T_\theta$ by optimizing the loss in Equation (14).

---

## K  EXTRA EXPERIMENTS

### K.1  DUELING BANDIT EXPERIMENTS

We consider dueling bandit (**DB**) problems with a shared linear bandit structure across varying DB problems Yue et al. (2012). All DB problems have the same action space $\mathcal{A}$, i.e., same number of bandits. To facilitate generalization to new DB problems, we assume a *fixed* bandit feature mapping $\phi : \mathcal{A} \to \mathbb{R}^d$ such that $\phi(a) \in \mathbb{R}^d$ is the feature for bandit $a$ shared by all DB problems. Each DB problem $\tau$ is characterized by a vector $\theta_\tau \in \mathbb{R}^d$ such that the expected reward of bandit $a$ for task $\tau$ is defined as $r_\tau(a) = \theta_\tau^\mathsf{T} \phi(a)$. To create challenging bandit problems with stochasticity, we assume that the observed reward is stochastic, that is, the observed reward after selecting $a$ in task $\tau$ follows a Gaussian distribution $r \sim \mathcal{N}(r_\tau(a), \kappa^2)$ where $\kappa^2 = 0.3$ represents variance of reward observations.

In a DB problem, at each time step $h \in [H]$ within a horizon $H$ , the agent chooses two actions $a, a'$ and receives a preference label $y$ on which of the two chosen actions is more preferred. Specifically, we follow the BT model to assume that $\mathbb{P}(y = 1|a, a', \tau) = \mathbb{P}(a \succ a|\tau) = \sigma(r_\tau(a) - r_\tau(a'))$. We underscore that $r_\tau$ is not observed in DB problems and we need to infer its information solely from the preference label $y$. The **goal** of DB problems is to find a bandit $a^\star \in \mathrm{argmax}_a \, r_\tau(a)$ that maximizes the expected reward. This is equivalent to a von Neumann winner which has probability at least 0.5 to be preferred over any bandits. We choose $|\mathcal{A}| = 20$, $d = 10$, and $H = 200$.

Note that the DB problems are special cases of our proposed **I-PRL** setting without state transitions. To this end, we evaluate our proposed **ICPO** framework on DB problems, as *it is designed for the I-PRL setting* and *its pretraining does not require the optimal bandit information*. The pretraining dataset for ICPO are generated as follows.

**Pretraining Dataset.** We first generate the bandit features $\phi(a) \sim\sim \mathcal{N}_d(0, I_d/d), \forall a \in \mathcal{A}$, independently following a Gaussian distribution. We generate $m$ DB problems in total and one context dataset for each generated DB problem. To this end, for each pretraining DB problem $\tau_i$, we sample its parameter $\theta_i$ independently from other problem parameters following $\theta^i \sim \mathcal{N}_d(0, I_d/d)$. To generate the context dataset $D^I$ for $\tau_i$, at each step $h$, we randomly sample a pair of distinct actions $(a_h, a'_h)$ following a uniform distribution over all pairs of different bandits. We do not enforce any extra coverage of the optimal bandits. We collect 100k context datasets for DB problems.

**Evaluation and Baselines.** To benchmark our ICPRL frameworks, we compare them to *Double Thompson Sampling* (**DTS**) Wu & Liu (2016), one of the most competitive DB algorithms. We deploy the pretrained TM policy $T_\theta$ and DTS to *new* DB problems. In the offline setting, we are given a context dataset $D^I$. Conditioned on $D^I$, our method ICPO follows the policy $T_\theta(a|D^I)$ to choose the first action $a \in \mathrm{argmax}_{\tilde{a}} T_\theta(\tilde{a}|D^I)$ and randomly sample the second action $a'$; DTS also utilizes the same bandit preferences in $D^I$ to select a pair of actions $(a, a')$, as detailed in Wu & Liu (2016). In terms of metrics, we follow the convention to use the *weak regret* defined

as follows. For a given pair of actions $a$ and $a'$, their weak regret $\mathrm{reg}$ for a DB problem $\tau$ is $\mathrm{reg}(a, a') = \min\left(R_\tau^\star - R_\tau(a), R_\tau^\star - R_\tau(a')\right)$, where $R_\tau^\star$ is the optimal expected reward for $\tau$. We evaluate both ICPO and DTS from three types of context data with different types of behavioral policies: (i) uniformly random behavioral policies; (ii) behavioral policies sampled from a Dirichlet distribution with uniform prior; (iii) DTS.

**Results Discussion.** We present key results in Figures 7. Note that the DB problems we consider are indeed challenging problems because the agent needs to identify the optimal bandit out of $|\mathcal{A}| = 20$ bandits within only $H = 200$ steps. This is verified in Figures 7, where DTS struggles to reduce its regret. Despite of these challenges, with all three context data, *our method ICPO significantly outperforms DTS*, showing considerably *faster regret decrease* and *consistently improving over the behavioral policies*. These results prove that our framework can pretrain transformers to efficiently solve new DB problems. In addition, the pretrained TM policies also demonstrate *robustness to distribution shift of context data*: although the pretraining data are generated by the uniformly random policies, ICPO consistently demonstrate strong performance even when the context data comes from Dirichlet behavioral policies or DTS.

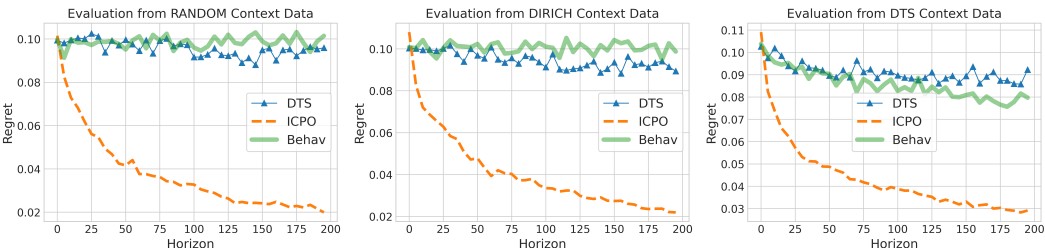

Figure 7: Performance Comparison for Dueling Bandit Problems. We evaluate performance under three types of context datasets, generated by different behavioral policies. **Random** context data is generated by a uniformly random behavioral policy. **Dirichlet** context data is generated by a behavioral policy sampled from the uniform Dirichlet distribution. **DTS** context data is generated by the competitive DB algorithm Double Thompson Sampling.

### K.2 MDP EXPERIMENTS

**T-PRL results in Darkroom.**

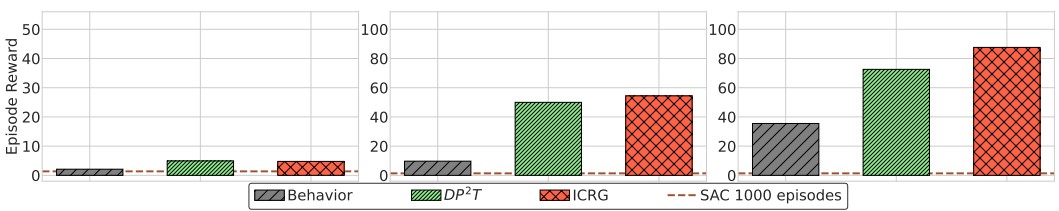

(a) Darkroom (T-PRL) **average episode reward** with context datasets of **low**, **medium**, and **high** quality.

## L IMPACT OF $\lambda$ FOR ICPO

In this section, we analyze the effect of the hyperparameter $\lambda$ on the episode rewards.

We conduct two different ablation experiments in DarkRoom and Meta-World to comprehensively understand the effect of $\lambda$. In Darkroom, we use $\lambda$ to scale the log-probability of *non-preferred* actions while in Meta-World we use $\lambda$ to scale the log-probability of *preferred* actions. When **scaling non-preferred actions**, smaller values of $\lambda$ (e.g., $\lambda < 1$) motivate the TM policy to focus more on *increasing the log-probability of preferred actions*. Thus, the model performance is expected to increase with when $\lambda$ decreases. In contrary, when **scaling preferred actions**, a small $\lambda$ (e.g., $\lambda < 1$) in fact motivates the TM policy to focus on *decreasing the log-probability of non-preferred actions*.

However, this does not guarantee to increase the log-probability of preferred actions. Thus, when using $\lambda$ to scale preferred actions, we should choose large values for $\lambda$, e.g., $\lambda > 1$.

This is exactly verified in Figure 9, as increasing $\lambda$ leads to opposite performance change in Dark-Room and Meta-World. When $\lambda$ is scaling non-preferred actions for DarkRoom, *increasing* $\lambda$ value *decreases* performance. In comparison, when $\lambda$ is scaling preferred actions for Meta-World, *increasing* $\lambda$ value *increases* performance. These results support our insights regarding the effect of $\lambda$. An open question remains: should preferred or non-preferred actions be scaled to achieve better generalization? We leave this investigation to future work.

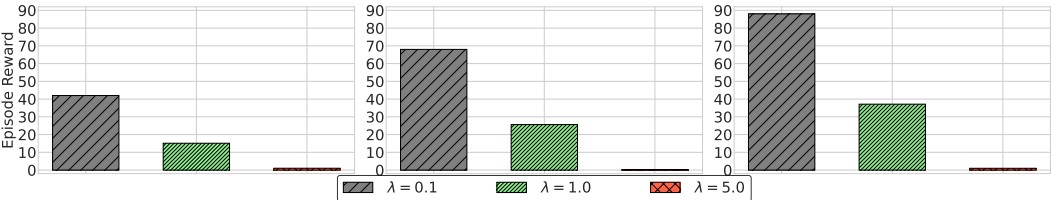

(a) Darkroom. ICPO under different $\lambda$ scaling the log-probability of *non-preferred* actions with context datasets of **low**, **medium**, and **high** quality.

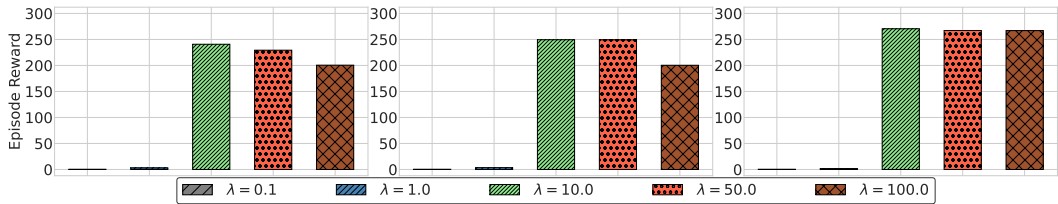

(b) Meta-World. ICPO under different $\lambda$ that scales the log-probability of *preferred* actions with context datasets of **low**, **medium**, and **high** quality.

Figure 9: Ablation Studies on the Effect of $\lambda$ for ICPO.

# M USE LLMS TO LABEL TRAJECTORY PREFERENCE

LLMs are powerful tools for general-purpose tasks. We leverage LLMs to generate preference labels for trajectories in **Darkroom**, demonstrating that our framework integrates seamlessly with LLMs to scale by reducing the cost of manual preference labeling. Figure 10 illustrates the prompts used for preference labeling. The trajectory states represent the sequence of grid positions visited by the agent, whereas the trajectory goal corresponds to a single state that the LLM leverages for evaluation. We test multiple open-source LLM modes for this purpose, including Qwen2.5-7B-Instruct, Qwen2.5-14B-Instruct, Qwen2.5-32B-Instruct and Qwen2.5-Math-7B-Instruct (Yang et al., 2025). Surprisingly, these models perform poorly on preference labeling in Darkroom, primarily due to their inability to accurately count the occurrences of goal states. However, this limitation can be readily addressed by leveraging tool-augmented LLMs. We verify that, under the same prompt, ChatGPT achieves 100% accuracy on preference labeling for 100 randomly sampled trajectories pairs, with the labels calculated through the underlying reward function as the true labels. Figure 11 illustrates a sample response from ChatGPT, which incorporates a Python snippet to facilitate preference labeling.

(a) System prompt used for preference labeling with an LLM.

(b) User prompt used for preference labeling with an LLM.

Figure 10: Prompt input to the LLM for preference labeling.

Figure 11: An example response from Chatgpt.

# N    PSEUDOCODES

# O    COMPUTATION RESOURCE

All experiments were run on 2 NVIDIA RTX A6000 GPUs (48 GB VRAM) mounted in a 64-core AMD Threadripper workstation with 256 GB RAM. Pretraining each transformer model took between

---

**Algorithm 2** ICRG Pipeline

---

1: **Input:** Pretraining dataset $\mathcal{D} = \{D_i = (\xi_i^+, \xi_i^-)\}$; reward model $R_\psi$, policy model $T_\theta^R$
2: //   Pretraining for the reward model
3: Randomly initialize $R_\psi$.
4: Pretrain $R_\psi$ with $\mathcal{D}$ by optimizing the objective function in Equation (10).
5: //   Label rewards with the pretrained reward model
6: Generate rewards for transitions in $\mathcal{D}$ with $R_\psi$ to have a pretraining dataset $\mathcal{D}^R$ containing trajectories with estimated rewards.
7: //   Pretraining for the policy model with DIT
8: Randomly initialize the TM policy $T_\theta^R$ that requires reward information.
9: Use the DIT framework (Algorithm 1) to pretrain $T_\theta^R$ with the reward-augmented pretraining dataset $\mathcal{D}^R$
10: //   Deployment
11: Upon deployment, receive a pair of preferred and non-preferred trajectories $D^T = (\xi^+, \xi^-)$ and a context dataset $D = \{s_1, a_1, \ldots, s_H, a_H\}$ without reward information.
12: Label all state-action pairs $(s, a)$ in $D$ with $R_\phi(s, a|D^T)$ to have an augmented context dataset $D^R = \{s_1, a_1, \widehat{r}_1, \ldots, s_H, a_H, \widehat{r}_H\}$.
13: Deployment the DIT pretrained TM policy $T_\theta^R(a|s, D^R)$.

---

6 and 12 hours, depending on the environment and dataset size. Each SAC policy was trained for approximately 3 hours per task. Across all runs (including pretraining, evaluation, SAC rollouts, and reward estimation), the total compute requirement was approximately 900 to 1100 GPU hours.

