# OpenReview forum: "Learning in Context, Guided by Choice:  A Reward-Free Paradigm for Reinforcement Learning with Transformers"
_ICLR.cc/2026/Conference — Submitted to ICLR 2026_

### Official Review · Reviewer_EAD1 · 2025-10-29

**Soundness:** 2
**Presentation:** 3
**Contribution:** 2
**Rating:** 2
**Confidence:** 4

**Summary:**

The paper presents a method for learning transformer-based control policies that rely on in-context learning abilities to generalize to new tasks and learn from offline data. The proposed approach ICPRL relies on conditioning the transformer policy on preference labelled trajectories both during pre-training and at deployment to encourage the policy to infer good versus bad behavior without requiring access to an explicit reward signal. After introducing this approach, the paper identifies that it does a poor job of learning from the preference data. Therefore, the paper introduces a modified approach that relies on a DPO-like objective when preference feedback is available per step and preference sample conditioned reward model when preference feedback is at the trajectory level. The results are presented for the per step and the trajectory level preference feedback in Darkroom (per step only) and MetaWorld. In some settings the proposed approach outperforms while in others it matches or falls short of the baselines.

**Strengths:**

- The paper applies the proposed approach to two preference feedback settings: at the step level and at the trajectory level.
- The paper is clear and easy to follow.
- The primary modification that is proposed is simple and not overly complex or convoluted.
- In MetaWorld, the ICRG results are stable across data quality conditions or increasing with data quality whereas other methods lack an consistent performance increase as data quality improves.

**Weaknesses:**

- Given the baselines provided, it is not easy to identify the benefits of the including preference labelled data in the reward nor policy's context. Examples of baselines that would help to make this clear include, training a reward model on the preference data and doing ICRG without the proposed preference dataset conditioning and using the proposed reward modelling approach to train a policy without conditioning on the preference samples.
- There are no results indicating how noisy preference labels impact reward and policy learning.
- The bulk of experiments and results are not in the main body of the paper. Instead 8/9 pages are dedicated to set up, despite the paper claiming the experiments "prove" the benefits of the proposed method. More of the experiments should be in the main body.
- The experiments are describing a proving the proposed method is generalizable. Experiments don't "prove", they demonstrate, show, suggest, or provide evidence.
- The contribution of the proposed "In-Context Preference Optimization" is not clear. It seems like an application of DPO.
- The paper presents its contributions as more complex than what they appear to be in practice. The main contributions seem to be applying DPO to per step preferences, learning a reward function conditioned on preference labelled examples, and doing exactly ICRL with the addition of conditioning on preference samples. The simplicity of this is hidden in the presentation.
- Instead of SAC, a meta-learning or multi-task algorithm that learns from the ground truth reward should be used.
- The experimental set up is not clearly detailed. For example, the number of tasks used for training versus evaluation is not clear nor which sets of tasks were used in each split.

**Questions:**

- What was the multi-task set up for MetaWorld? Which tasks were trained on? Which were evaluated on?
- Why are you reporting episode reward instead of success rate for MetaWorld?
- How many random seeds were used?
- Are the results in Figure 2 and Figure 3 means over tasks? If so, what is standard error from the mean? If not, what were the episode rewards computed over?
- It is said that the context datasets are of low, medium, or high quality. What does this mean when the context dataset are preference samples? Does the quality description cover the trajectories or the label quality? If label quality, how was that manipulated?

---

> ### Author Response · Authors · 2025-12-03
>
> We sincerely thank the reviewer for the time and effort dedicated to evaluating our manuscript. We hope that the following response will clarify the raised points and address the concerns constructively.
>
> ### Presentation and Clarification on Contributions.
>
> > **Main Content.** ```The bulk of the experiment and results are not in the main body of the paper ...```
>
> Our contribution is to **extend in-context RL to preference-based settings**, showing that **transformers can directly use preference signals as contextual information for in-context adaptation**.
>
> As a result, due to space constraints, **the main paper focuses on formulating the problem, formalizing the paradigm, and presenting theoretical insights**. We agree that the structure can be improved and will reorganize the manuscript to surface more experimental results in the main body.
>
> > **Contributions.** ```The contribution of the proposed "In-Context Preference Optimization" is not clear...The simplicity of this is hidden in the presentation.```
>
> As detailed above, this work proposes **ICRL with only preference labels**, extending ICRL to **challenging yet common** scenarios where immediate rewards are difficult to design or obtain. For comprehensiveness, we propose two settings with step-wise preference label (**I-PRL** and **T-PRL**). Our ICPO algorithm is proposed to efficiently solve the **I-PRL** setting.
>
> While it shares some motivation with DPO ( i.e., to avoid learning a separate reward function, as we already cited and discussed in our manuscript), **ICPO is designed for a completely different problem**. We humbly believe that the value of ICPO should **not** be criticized just due to its share of motivation. **It is more than common for machine learning researchers to  be motivated by ideas from different fields when solving new problems**. In addition, our algorithm ICAG designed for the **T-RPL** setting is also novel and orthogonal to DPO, providing notable technical contribution.
>
> We are **glad** that the reviewer finds simplicity in our methods. This is **exactly** what we want to achieve: **we propose an important yet challenging problem, and we address it with our insights leading to a simple yet effective solution**.
>
> ### Clarification of the Methods.
>
> > **Multiple Components.** ``` ```
>
> We would like to emphasize that in T-PRL, the reward estimator and policy module are independent components. The reward estimator generates per-step reward signals, and the in-context policy consumes a *single* trajectory input (as opposed to paired preferences) to produce the desired actions. Building on previous evidence that in-context policy learning is effective, our contribution is to show that this mechanism can be directly leveraged in preference-based reinforcement learning as well.
>
> > **Noisy Preference Labels.** ```There are no results indicating how noisy preference labels impact ...```
>
> The preference labels in our experiments are generated by sampling from a softmax distribution over the summed returns of the two trajectories, rather than deterministically selecting the higher-reward one. **This stochastic labeling naturally introduces noise**, making the preference supervision more realistic and less idealized.
>
> > **SAC Baseline.** ```Instead of SAC, a meta-learning or multi-task algorithm that learns ...```
>
> We do not train SAC in a multi-task adaptation setting; instead, we train it on each task independently under a budget to emulate a learning-from-scratch  scenario. This setup provides the strongest attainable performance for SAC on each **unseen** task when given a **limited** number of interactions (the targeted scenario of ICRL), and thereby highlights the **inherent difficulty of the tasks**.
>
> To prove the meta-learning ability of our methods, we compare with **DPT** with access to both optimal action labels and immediate rewards. This is one of the **strongest** baselines for ICRL. We prove the efficacy of our methods by showing our methods' performances are close to those of DPT despite **without access** to that information.

---

> ### Author Response · Authors · 2025-12-03
>
> ### Experiments.
>
> > **Experiment Details.**
>
> MetaWorld trains on 45 tasks and evaluates on 5 held-out tasks, whereas Darkroom uses 80 training tasks and 20 test tasks. MetaWorld allows specifying goal positions, which are treated as distinct tasks; we follow the standard protocol by using the predefined 45 tasks for training and 5 for testing. For each task, we run 5 random seeds and evaluate over 10 test episodes, reporting the mean performance across checkpoints from these seeds.
>
> > **Context Quality.** ``` It is said that the context datasets are of low, medium or high quality ...```
>
> In our experiments, context quality refers specifically to the quality of the trajectories provided to the policy as contextual input. The preference context used by the reward estimator to generate rewards remains fixed across all experiments.
>
> Quantifying the quality of perference pairs is challenging, and to the best of our knowledge, there is no universally accepted metic. Yet, an inituition may exist that trajectory diversity may play an important role. When the two trajectories in a pair differ significiantly---e.g., one is success, the other a failure---this yields:
>
> * **Lower Label Noise.**  Bradley-Terry model results in a lower probability of incorrect labels with larger reward gaps.
> * **Broader state-action coverage.** It helps the model to better understand a target task.
>
> When using a **single behavior policy** to collect a preference pair, it would be thus reasonable to consider that a mid-performaing policy tend to generate better pairs than low or high policies. This is because that the policy naturally generates both successful and failed trajectories, which are more likely to provide more diverse and informative comparisons.
>
> To validate this point, we conducted an ablation in which we varied both the context quality provided to the in-context policy and the context quality provided to the reward estimator. The table below summarizes ICRG's performace under different combinations:
>
>
> | Policy Context\\ Reward Estimator context | Low   | Mid       | High  |
> | ------------------------------------------ | ----- | --------- | ----- |
> | **Low**                                    | 149.7 | **181.4** | 160.7 |
> | **Mid**                                    | 151.6 | **187.9** | 167.2 |
>
> We observe that preference pairs generated by the mid-performing policy consistently yield higher returns, which **supports our hypothesis**.
>
> > **Performance Metrics.** ```Why are you reporting episode reward instead of success rate for MetaWorld?```
>
> We follow the standard ICRL evaluation protocol, which uses episode reward as the primary metric because it reflects both task completion and control quality. **Success rate can be derived from reward thresholds**, but reward provides a **more fine-grained** and stable signal across variations, which is why we report it here.

---

### Official Review · Reviewer_ejrF · 2025-11-01

**Soundness:** 3
**Presentation:** 3
**Contribution:** 2
**Rating:** 6
**Confidence:** 3

**Summary:**

This paper explores extending in-context reinforcement learning (ICRL) with supervised pretraining on preference data instead of explicit reward signals. The authors provide methods for learning from both per-step preference and trajectory-wise preference. The experiments show the proposed method performs on par with or exceeds baselines that use explicit reward signals.

**Strengths:**

- The writing is clear and easy to understand.
- The proposed methods are novel extensions of the existing ICRL frameworks to the preferential data domain.
- The proposed methods remain competitive in the absence of explicit reward signals.

**Weaknesses:**

- It seems each experiment only contains one run, which raises concerns about statistical rigour.
- I am concerned about the practicality and robustness of the in-context reward estimation and reward relabelling approach. It involves two stages of in-context learning: one for learning the in-context reward estimator and the other for learning the in-context policy.

**Questions:**

- Is the in-context policy learning independent from the in-context reward estimator learning, or does it use the relabelled data by the estimator for training?
- Why do the authors choose a regularized objective in this case? If the ultimate goal is to maximize returns, I don't see why one should penalize deviations from the behaviour policy.

---

> ### Author Response · Authors · 2025-12-03
>
> We would like to first appreciate the reviewer's time and effort devoted to our manuscript. We hope that the following response could resolve your concerns.
>
> ### Clarification of the Methods.
>
> > **Experiment.** ```It seems each experiment only contains one run ...```
>
> For all plots in the manuscript, we report results **averaged across test tasks, 10 evaluation episodes per task, and checkpoints from five random seeds**. The presented curves therefore reflect the **mean performance** of the proposed method. We thank the reviewer for noting this potential source of confusion, and we will include standard deviation in the next revision of the manuscript to improve clarity.
>
> > **In Context Reward Esimator.** ```Is the in-context policy learning independent from ...```
>
> The in-context policy learning procedure leverages rewards predicted by the in-context reward estimator. Our key insight is to introduce only the minimal components required for in-context reinforcement learning to operate effectively in a preference-based setting. It is thus the case that the reward estimator is used to generate reward labels in-context, which are then directly used to guide policy learning.
>
> ### Other Points.
>
> > **Practicality.** ``` I am concerned about the practicality and robustness ...```
>
> Our method illustrates that in-context learning can be directly leveraged for preference learning, enabling an independent reward-labeling mechanism that functions separately from downstream policy modules. Once rewards are generated, they can be seamlessly integrated (or replaced) in various downstream pipelines. We further show through theoretical derivation and empirical evidence that the resulting reward estimates are stable and reliable.
>
> > **Regularization.** ```Why do the authors choose a regularized objective ...```
>
> When learning from an offline dataset and estimate value functions from it, it's always beneficial to regularize the policy not too far from the behavior policy, such that the value estimation could be accurate enough to be meaningful. A similar thought has been demonstrated through [1].
>
> [1]Fujimoto, S., Meger, D., & Precup, D. (2019). *Off-Policy Deep Reinforcement Learning without Exploration.* arXiv:1812.02900.

---

### Official Review · Reviewer_8ssj · 2025-11-03

**Soundness:** 2
**Presentation:** 3
**Contribution:** 1
**Rating:** 2
**Confidence:** 4

**Summary:**

This paper presents in-context preference reinforcement learning (ICPRL), a form of multi-task reinforcement learning where the adaptation to different tasks is done in-context (ie. changing the inputs to the transformer model), and the reward is derived (implicitly or explicitly) from binary preferences.

The paper proposes two implementations of ICPRL: in the first one the preferences are considered over state-action pairs (I-PRL), in the second one the preferences are considered over trajectories (T-PRL). Additionally, three pertaining strategies for I-PRL and T-PRL are explored: 1) DP2T where the preferences are simply provided in context, 2) ICPO (only for I-PRL) where preferences are used to directly optimise the transformer model (analogously to Direct Preference Optimisation), and 3) ICRG (only for T-PRL) where a multi-task reward function is learnt from trajectory preferences.

Experiments with MetaWorld and DarkRoom show that I-PRL+ICPO and T-PRL+ICRG are competitive with the ICRL baseline, but without the need for explicit reward functions or high-quality reward transitions.

**Strengths:**

* The paper is well written, provides a good motivation, and is easy to follow (though see nitpicks below).
* The derivation of ICPO is very interesting and so is its connection to DPO.
* The paper tackles an challenging problem (multi-task RL) through creative means (mixing in-context learning with preference-based reinforcement learning)
* The theoretical derivation of the paper is _very_ thorough with many useful appendices (though note I did not have time to review all of the appendices).
* The paper explores thoroughly two very different approaches to in-context preference reinforcement learning: I-PRL and T-PRL.

**Weaknesses:**

* **W1**: The form of multi-task learning employed in the experiments is too narrow. For both DarkRoom and MetaWorld, "multi-task" boils down to different initial and goal states. Indeed, it was precisely this issue that motivated MetaWorld [1] (quoting from the abstract, emphasis mine):

> However, much of the current research on meta-reinforcement learning focuses on task distributions that are very narrow. For example, a commonly used meta-reinforcement learning benchmark uses different running velocities for a simulated robot as different tasks. **When policies are meta-trained on such narrow task distributions, they cannot possibly generalize to more quickly acquire entirely new tasks**. Therefore, if the aim of these methods is enable faster acquisition of entirely new behaviors, we **must evaluate them on task distributions that are sufficiently broad** to enable generalization to new behaviors.

The current experiments on MetaWorld are a necessary baseline, but the impact of ICPRL would be much more significant if the method worked well on the MT-10 portion of MetaWorld tasks.

* **W2**:  Though the related works section is really well redacted, it is missing a discussion of how ICPRL differs from goal-conditioned reinforcement learning. Similarly, connections to meta-learning should be explored.

* **W3**: Some key ablations are missing, specifically: the number of preferences used during pre-training, the number of examples given in-context, and the accuracy of the learnt reward function for ICRG (see questions below for more details).

* **W4**: The effect of human-gathered preferences (say for T-PRL) is not studied. Without such a study the applicability of ICPRL is reduced to settings where preferences may be automatically obtained.
---------

[1] Yu et al. (2019) "Meta-World: A Benchmark and Evaluation for Multi-Task and Meta Reinforcement Learning" CoRL.

**Questions:**

### Questions (in no particular order)

* **Q1**: The paper states that AD (Laskin et al. 2002) "assume $D^R = \\left\\{\\epsilon_{i,j}\\right\\}^J_{j=1}$, a set of trajectories  $\\epsilon_{i,j}$ collected by increasingly improving policies", yet the reference cited seem to provide state-action preferences instead. Did I miss something in Laskin et al? If not, can you provide an example of an ICRL method in the literature that provides trajectory-level rewards?
* **Q2**: For I-PRL how are the in-context state-action pairs sampled? Are they sampled independently? Or do they follow the state-action pairs of the preferred trajectory?
* **Q3**: Are there any alternatives to the uniform policy for ICPO (line 345)? Could you really on a policy that maximises exploration of the state-space for instance?
* **Q4**: For T-PRL+DP2T, is the binary preference inserted before or after the trajectories? If it's inserted after, is the preference always on the same position (ie are the trajectories always of the same length?)
* **Q5**: For Figs 2 & 3, are these the results of a single run? What is the spread across different initialisations for the same task (same initial and goal states)? What is the spread across different tasks?
* **Q6**: What is the effect of increasing/decreasing the size of the pre-training dataset on final performance?
* **Q7**: How many examples are given in-context? What is the effect of increasing/decreasing in-context samples on final performance?
* **Q8**: In Fig 2 (bottom, MetaWorld), why does increasing the quality of the trajectories result in a decrease in performance for I-PRL+DP2T?
* **Q9**: in Fig 2 (bottom, MetaWorld) the performance of SAC looks poor. Did SAC actually converge during training?
* **Q10**: what are the effects of varying the numbers of tasks for use in pertaining/testing? How did you come up with the current splits (for DarkRoom the ratio is 80/20, whereas for MetaWorld is 90/10).
* **Q11**: related to the previous question and following up on lines 216-220, how many additional preference pairs are needed to get bring ICPRL to work a new task?

-------

### Nitpicks (do not affect rating, no need to follow up during rebuttal)

* **N1**: Appendix B has repeated sections from the main paper verbatim. Please include only what was not added in the main text (and ideally try to fit the literature review in the main paper).
* **N2**: Line 143: instead of "classification-like" I suggest "cross-entropy where the optimal action $a^\\star$ has $p=1$.
* **N3**: Are you certain that the reliability of labels can be _guaranteed_ by any LLM method? Do you have citation where this statement is proved?
* **N4**:  I would merge sections 5 and 6, since DP2T is just another pertaining strategy.

---

> ### Author Response · Authors · 2025-12-03
>
> We would like to thank the reviewer for their time and careful assessment of our manuscript. The raised questions are insightful and have been valuable in helping us clarify several points. We hope that our responses effectively resolve the concerns.
>
> ### Presentation and Others.
>
> > **Presentation.**
>
> Thank you for recognizing the writing quality of our work. We will incorporate the suggested revisions into the next iteration of the manuscript.
>
> > **LLM-Generated Preference Label.**  ```N3: Are you certain that the reliability of labels can be ... ```
>
> There is an existing line of work that leverages large foundation models for preference labeling, such as [1] and [2]. These works apply foundation models to the complex domain of robotics for preference labeling, and we expect the approach to generalize to a broader range of applications.
>
> ### Connection to Other Works.
>
> > **Connection to Algorithm Distillation (AD).** ```Q1: The paper state that AD assumes ... ```
>
> AD operates **with immediate rewards** and thus **without preference labels**. Each trajectory is fed into the transformer independently, rather than being evaluated or compared pairwise, and therefore no preference ordering or ranking is established. To the best of our knowledge, our work is the first to extend ICRL to a preference-based setting.
>
> > **Connection to Goal-Conditioned Reinforcement Learning.** ```W2: Though the related works section is really well redacted ...```
>
> Goal-conditioned RL typically assumes that a goal is **explicitly provided** to the policy as part of its input. In contrast, **ICRL infers the goal implicitly** from the provided trajectory without requiring it to be specified. In the in-context preference-based setting, the goal must be inferred even more indirectly through comparison between pairs of preference trajectories.
>
> ### Clarification of the Proposed Methods.
>
> > **Answer to Q2.** ```Q2: For I-PRL how are the in-context state-action pairs sampled? Are they sampled independently? Or do they follow the state-action pairs of the preferred trajectory?```
>
> We detailed the data generating process for I-PRL in Section 4.1 **I-PRL Context**.
>
> > **Answer to Q3.** ```Q3: Are there any alternatives to the uniform policy for ICPO (line 345)? Could you really on a policy that maximises exploration of the state-space for instance?```
>
> Thank you for this suggestion. One can absolutely choose policies other than the uniform policy. **In Section 6.1, we explained in detail our rationale** for choosing the uniform policy. Meanwhile, uniform policy should already be the policy with **maximum exploration** (there is **no** exploitation at all).
>
> > **Answer to Q4.** ```Q4: For T-PRL+DP2T, is the binary preference inserted before ...```
>
> For Implementation, we don't explicitly inject binary signals, but embed preferred and non-preferred trajectories into two embeddings and concatenate them together as a way to inject preference signals. Figure 5 in appendix illustrates the design.
>
> > **Answer to Q5.**  ```Q5: For Figs 2&3, are these the results of ...```
>
> We reported average results over 5 seeds and multiple mid-point checkpoints and all testing tasks with 10 episodes for the final plots. We will update the plots to include variance both within individual tasks and across tasks in the next revision of the manuscript.
>
> > **Answer to Q6.** ```Q6:What is the effect of increasing/decreasing the size of the pre-training dataset on final performance?```
>
> A larger pretraining data size would increase the performance of a preference-based model. We report the following results on DP2T as a verification.
>
>
> | Number of Preference Pairs (Pretraining) | 50000  | 30000  | 10000  | 5000   |
> | ---------------------------------------- | ------ | ------ | ------ | ------ |
> | **Mean**                                 | 91.785 | 67.350 | 51.195 | 45.170 |
> | **Standard Deviation**                   | 5.280  | 16.980 | 20.800 | 20.770 |
>
> > **Answer to Q7**. ```Q7: How many examples are given ...```
>
> We provide only a single pair of trajectories to the pretrained transformer for both training and deployment. This represents the most challenging setting, as additional context is known to generally improve performance, a trend well-established in prior literature.
>
> > **Answer to Q8**. ```Q8: In Fig 2 (bottom, MetaWorld), why does increasing the quality of the trajectories result in a decrease in performance for I-PRL+DP2T?```
>
> We believe this is because of the **complex nature** of MetaWorld. In this case, even high-reward trajectories cannot provide reliable high-quality step-wise labels. We need zoom out to consider the quality of trajectories in their **entirety**, i.e., using the trajectory-wise preferences. This is confirmed by our experiments of **T-PRL** (which uses trajectory preferences)  in Meta World, showing **consistent performance improvements with higher quality data**.

---

> ### Author Response · Authors · 2025-12-03
>
> > **Answer to Q9.** ```Q9: In Fig2, the performance of SAC looks poor ...```
>
> This behavior is utterly **expected**. The performance of  SAC algorithm is meant to **illustrate the difficulty of the testing tasks** from a single-task learning perspective. Specifically, SAC is trained from scratch on each task independently and is given **1000** trajectory budget. Our objective here is not to showcase the best performance SAC can achieve, but rather to establish a baseline that reflects how challenging these testing tasks are **without **ICRL models.
>
> > **Answer to Q10.** ```Q10: What are the effects of varying the number of tasks for use ...```
>
> There is a line of work analyzing the level of task diversity required for in-context learning to emerge, including [4] and [5]. These studies show that a transformer must be pretrained on tasks exceeding a certain diversity threshold for robust in-context learning behavior to emerge, enabling strong performance on unseen tasks.
>
> For verifying this conclusion on our setting, we vary the split of tasks used for training and test on Darkroom and provide the results of ICRG summarized in the following table.
>
>
> | Algorithm\\ Task Split (Trainig:Test)  | 90:10 | 60:40 | 40:60 |
> | ---------------------------------------- | ----- | ----- | ----- |
> | **ICRG**                                 | 89.32 | 78.65 | 46.27 |
>
> The performance gap between the second and third columns is substantially larger than the gap between the first and second columns, corresponing to the conclusion from prior works.
>
> In our experiments, we follow the experiment protocal from [6] for the specific task splits.
>
> ### Clarification of Contribution.
>
> Our main contribution is to **extend in-context RL to preference-based settings**, showing that **transformers can directly use preference signals as contextual information for in-context adaptation**. Our key insight is to reduce in-context preference-based RL to standard in-context RL, which only requires a straightforward derivation of per-step rewards. We aim to provide a thorough derivation of this reduction and to experiment with multiple settings that instantiate it.
>
> ---
>
> **References.**
>
> [1]Y. Wang, Z. Sun, J. Zhang, et al. RL-VLM-F: Reinforcement Learning from Vision Language Foundation Model Feedback. arXiv:2402.03681, 2024.
> [2]Yu, Chao, Qixin Tan, Hong Lu, et al. ICPL: Few-shot In-context Preference Learning via LLMs. arXiv preprint arXiv:2410.17233, 2024.
> [3]von Oswald, Johannes, Eyvind Niklasson, Ettore Randazzo, João Sacramento, Alexander Mordvintsev, et al.
> Transformers learn in-context by gradient descent. arXiv preprint arXiv:2212.07677, 2022.
> [4]Raventós, A., Paul, M., Chen, F., & Ganguli, S. Pretraining task diversity and the emergence of non-Bayesian in-context learning for regression. arXiv preprint arXiv:2306.15063.
> [5]Wu, J., Zou, D., Chen, Z., Braverman, V., Gu, Q., & Bartlett, P. L. How Many Pretraining Tasks Are Needed for In-Context Learning of Linear Regression? arXiv preprint arXiv:2310.08391.
> [6]Lee, J. N., Xie, A., Pacchiano, A., Chandak, Y., Finn, C., Nachum, O., & Brunskill, E.  Supervised Pretraining Can Learn In-Context Reinforcement Learning. arXiv:2306.14892.

---

### Meta-Review · Area_Chair_2C6j · 2026-01-05

**Summary:**

Most reviewers concern about the limited empirical experiments, which I agree. Hence, I recommend for rejection and encourage authors to consider suggestions and include more experiments.

**Reviewer Concerns:**

Most reviewers found the paper well-written and the problem setting interesting but raised significant concerns about empirical rigor and baselines: (1). Reviewer 8ssj argued the multi-task setup was too narrow (just changing start/goal states in MetaWorld/DarkRoom); (2). Reviewer EAD1 requested stronger baselines to isolate the benefit of preference conditioning, such as training a reward model on preferences without the proposed conditioning or using standard meta-learning/multi-task algorithms instead of single-task SAC; (3). Reviewers noted missing data on the number of preferences used, context size, and the impact of noisy preference labels. (4) Reviewer EAD1 questioned reporting episode reward instead of success rate for MetaWorld.

Reviewer EAD1 also also questions the novelty and contribution.

After reading the rebuttal where a few new results are provided, I agree with reviewers that the experiment evaluation is not complete and comprehensive.

**Reviewer Scores:**

I believe most reviewers shall maintain the scores.

---

### Decision · Program_Chairs · 2026-01-26

Reject